# CARD8 inflammasome activation triggers pyroptosis in human T cells

Andreas Linder[1,2], Stefan Bauernfried[1], Yiming Cheng[1], Manuel Albanese[1,3], Christophe Jung[1], Oliver T Keppler[1,3] & Veit Hornung[1,*]

## Abstract

Inflammasomes execute a unique type of cell death known as pyroptosis. Mostly characterized in myeloid cells, caspase-1 activation downstream of an inflammasome sensor results in the cleavage and activation of gasdermin D (GSDMD), which then forms a lytic pore in the plasma membrane. Recently, CARD8 was identified as a novel inflammasome sensor that triggers pyroptosis in myeloid leukemia cells upon inhibition of dipeptidyl-peptidases (DPP). Here, we show that blocking DPPs using Val-boroPro triggers a lytic form of cell death in primary human CD4 and CD8 T cells, while other prototypical inflammasome stimuli were not active. This cell death displays morphological and biochemical hallmarks of pyroptosis. By genetically dissecting candidate components in primary T cells, we identify this response to be dependent on the CARD8-caspase-1-GSDMD axis. Moreover, DPP9 constitutes the relevant DPP restraining CARD8 activation. Interestingly, this CARD8-induced pyroptosis pathway can only be engaged in resting, but not in activated T cells. Altogether, these results broaden the relevance of inflammasome signaling and associated pyroptotic cell death to T cells, central players of the adaptive immune system.

**Keywords** CARD8; inflammasome; pyroptosis; T cell; Val-boroPro
**Subject Category** Immunology
**The EMBO Journal (2020) 39: e105071**

## Introduction

Inflammasomes function as cytosolic sensors of microbial infection as well as perturbation of cellular integrity. A variety of inflammasome sensor pathways have been identified that converge on the activation of caspase-1, which plays an important effector function in antimicrobial defense and sterile inflammatory responses (Broz & Dixit, 2016). Currently, known inflammasomes include the NLRP1, NLRP3, NAIP/NLRC4, NLRP6, NLRP12, AIM2, and the PYRIN inflammasome. NOD-like receptors (NLR)-based inflammasome sensors typically display a three-domain structure: an N-terminal homotypic interaction domain that is part of the death-fold domain family (e.g. CARD or a pyrin domain), a central nucleotide binding and oligomerization domain, and a series of C-terminal leucine-rich repeats (LRRs). NLRP1 varies from this canonical architecture, as it harbors an additional C-terminal extension (see below). AIM2 and PYRIN are not part of the NLR protein family, yet also rely on an N-terminal pyrin domain for signal transduction. The activation of these inflammasome sensors results in the formation of a platform that allows the homotypic interaction domains to recruit the universal adapter protein ASC or caspase-1 directly. Direct or ASC-dependent caspase-1 recruitment then facilitates its autoproteolytic activation. Beyond serving as a simple signaling adapter, ASC recruitment typically results in the assembly of large filamentous structures that serve to strongly amplify caspase-1 activation (Broz & Dixit, 2016).

The NAIP/NLRC4 inflammasome directly senses components of the bacterial type III secretion system (T3SS) or flagellin, with NAIP serving as the direct receptor in this setting. While different NAIP proteins with varying ligand specificities exist in mice, the human system appears to encode for only one NAIP protein that is especially sensitive to T3SS needle proteins (Yang *et al*, 2013). Ligand binding by NAIP initiates the formation of a large oligomeric complex, in which the ligand-bound NAIP protein recruits NLRC4, causing a conformational change in NLRC4 which facilitates recruitment of other NLRC4 molecules. This results in the formation of a wheel-like structure, in which the N-terminal CARD domains of NLRC4 are solvent exposed, producing a caspase-1 recruitment platform.

Dissimilar to the NAIP/NLRC4 inflammasome, NLRP3 depends on the adaptor protein ASC to activate caspase-1, although ASC can enhance the NAIP/NLRC4 signal (Broz *et al*, 2010). NLRP3 also differs in that it is activated by a broad array of different stimuli, many of which converge on the induction of potassium ($K^+$) efflux along its physiological transmembrane gradient. This phenomenon is typically associated with the loss of membrane integrity in the context of cellular damage or certain forms of programmed cell death (Gaidt & Hornung, 2018). Given the low specificity of this process, NLRP3 is often engaged in the context of acute or chronic sterile inflammatory conditions. Analogous to NAIP/NLRC4, it is believed that NLRP3 assembles a wheel-shaped oligomer that serves as a seed to initiate inflammasome formation, yet structural insight is currently missing.

1 Gene Center and Department of Biochemistry, Ludwig-Maximilians-Universität München, Munich, Germany
2 Department of Medicine II, University Hospital, Ludwig-Maximilians-Universität München, Munich, Germany
3 Max von Pettenkofer Institute, Virology, Ludwig-Maximilians-Universität München, Munich, Germany
*Corresponding author. Tel: +49 (0) 89 2180 71110; E-mail: hornung@genzentrum.lmu.de

The domain architecture of NLRP1 is homologous to other inflammasome-forming NLR proteins, while its C-terminus is extended with two additional domains: a FIIND (function to find domain) and a CARD. Its FIIND constitutes an autoproteolytic domain that leads to NLRP1 processing into an N- and C-terminal part that remain non-covalently associated with one another (Mitchell *et al*, 2019). Of note, this autoprocessing event is critical to ascertain NLPR1's functionality. Most mechanistic studies on NLRP1 activation have been carried out in the murine system. Here, *B. anthracis* infection has been identified to trigger inflammasome activation in an Nlrp1b—one of several Nlrp1 paralogs present in the murine system—dependent manner. Mechanistically, the *B. anthracis*-encoded lethal factor (LF) cleaves murine Nlrp1b close to its N-terminal region. This cleavage event exposes a neo-N-terminus of the large Nlrp1b fragment, which then constitutes an N-degron signal that is detected by E3 ubiquitin ligases of the N-end rule pathway, marking it for proteasomal degradation (Chui *et al*, 2019; Sandstrom *et al*, 2019). Removal of this fragment, however, allows the release of the non-covalently associated C-terminal portion of Nlrp1b, distal to the autoprocessed FIIND. This thereby-released fragment contains the signal-transducing CARD that then recruits and triggers caspase-1 activation. In addition to the cleavage-induced exposure of an N-degron and the associated detection by endogenous E3 ligases of the N-end rule pathway, direct modification of Nlrp1b by pathogen-encoded E3-ligases has also been identified to activate Nlrp1b (Sandstrom *et al*, 2019). In light of the pivotal role of proteasomal degradation in this response pathway, this mechanism of Nlrp1b activation has also been referred to as "functional degradation" (Sandstrom *et al*, 2019). Of note, the N-terminal regions of murine Nlrp1b and human NLRP1 are poorly conserved, and human NLRP1 lacks the cleavage site for anthrax toxin lethal factor. As such, it remains to be determined what microbial pathogen triggers the activation of the human NLRP1 inflammasome.

Another mode of murine and human NLRP1 activation has been identified by characterizing the cell death-inducing activity of the non-selective DPP-inhibitor Val-boroPro (VbP, Talabostat) and related compounds. VbP is an inhibitor of post-proline-cleaving serine proteases that include FAP, DPP4, DPP7, DPP8, and DPP9. In various human and murine cell lines, VbP was found to trigger caspase-1-mediated pyroptosis (Okondo *et al*, 2017; Taabazuing *et al*, 2017). This was later identified to depend on Nlrp1b in murine macrophages (Okondo *et al*, 2018), as well as on NLRP1 in human keratinocytes (Zhong *et al*, 2018). Moreover, complementing NLRP1-deficient cell lines with NLRP1 or Nlrp1b expression constructs rendered these cells competent to VbP-induced pyroptosis. Interestingly, in the course of these studies it was also found that VbP treatment could engage CARD8 to trigger caspase-1 activation (Johnson *et al*, 2018; Zhong *et al*, 2018). CARD8 is structurally related to NLRP1, in that it is homologous to the C-terminal portion of NLRP1, consisting of a small N-terminal region, a FIIND, and a CARD domain. While present in primates, CARD8 is not found in the murine system. Indeed, in human myeloid cells, differently from human keratinocytes, CARD8 but not NLRP1 is responsible for pyroptosis induction upon VbP treatment (Johnson *et al*, 2018). The mechanism by which VbP triggers NLRP1 or CARD8 activation is not fully understood. Inferring from the activity of more specific inhibitors and genetic perturbation studies, it could be concluded that DPP8 and DPP9 inhibition, but not other DPPs, function upstream of pyroptosis (Okondo *et al*, 2017). While both enzymes have been shown to restrain NLRP1/CARD8 inflammasome activity, it appears that DPP9 seems to play a more dominant role in this activity (Okondo *et al*, 2017). Apart from functional data, binding studies have shown that DPP8 and DPP9 bind to NLRP1 and CARD8 (Zhong *et al*, 2018; Griswold *et al*, 2019) and based on these experiments it had been suggested that direct interaction of DPP8 or DPP9 with these sensors restrains their activity, while inhibition of their activity relieves this inhibition. However, while DPP9 inhibition disrupts the binding of DPP9 to NLRP1, the interaction of CARD8 and DPP9 is not affected by DPP9 inhibition. Instead, it appears that the inhibition of its catalytic activity, rather than its binding is of relevance for inflammasome activation (Griswold *et al*, 2019). As such, the exact mechanism of how the activity of DPP8 and DPP9 functions upstream of CARD8 or NLRP1 remains enigmatic. Nevertheless, it is interesting to note that VbP-induced CARD8 or NLRP1 activation is also completely dependent on a step that involves proteasomal degradation.

Upon autoproteolytic activation, caspase-1 cleaves a select number of substrates that include the highly pro-inflammatory cytokine IL-1β as well as the pore-forming molecule GSDMD. Caspase-1-dependent GSDMD cleavage at Asp275 relieves the auto-inhibitory control of the C-terminal portion of GSDMD on its N-terminus. This allows the recruitment of the N-terminal fragment to the inner leaflet of the plasma membrane, which results in its cooperative assembly and pore formation. The GSDMD pore allows macromolecules such as IL-1β to pass through the membrane, yet it also dissipates the electrochemical ion gradients across the membrane allowing for uncontrolled water influx. When subject to additional mechanical forces, such destabilized cells can rupture and release further cytosolic content (Davis *et al*, 2019). This lytic form of cell death plays an important role in the passive release of IL-1β and potentially other pro-inflammatory mediators (Broz *et al*, 2020).

Previous studies have shown that inflammasome components can play important roles in T-cell biology. From a mechanistic standpoint, two independent roles can be delineated: First, in line with their well-described role in myeloid or epithelial cells, inflammasome functionalities associated with caspase activation have been described for T cells. As such, it has been shown that resting tonsillar T cells undergo pyroptosis upon non-productive HIV infection (Doitsh *et al*, 2014). This phenomenon was then later attributed to IFI16 sensing HIV-derived nucleic acids, triggering the activation of an inflammasome complex (Monroe *et al*, 2014). Another study has found that T cell-intrinsic engagement of the NLRP3 inflammasome critically contributed to IFNγ production and Th1 differentiation (Arbore *et al*, 2016). Moreover, it has been shown that an NLPR3-ASC-CASP8 signaling axis exerts the maturation and release of IL-1β in Th17 cells, thus contributing to disease pathology in a model of experimental autoimmune encephalomyelitis (Martin *et al*, 2016). While these studies provided unconventional models for the engagement of inflammasome components (e.g. IFI16) or associated downstream biological effects (e.g. caspase-8), they are in line with the concept of caspase-1/8 exerting proteolytic activity downstream of inflammasome activation. An alternative second line of research proposes that inflammasome components or caspase-1 itself exert functions in T cells beyond their conventional roles. To this end, it has been shown that NLRP3 functions as a transcriptional regulator of Th2 differentiation

as evidenced in a loss-of-function (Bruchard *et al*, 2015) or gain-of-function setting (Braga *et al*, 2019). Adding to this, a recent study suggests that caspase-1 plays a critical role in Th17 differentiation, independent of its catalytic activity (Gao *et al*, 2020). While it is difficult to reconcile these different studies into a plausible working model, it has to be considered that differences in cell type, ligands, species, and experimental readouts may account for these results. Nevertheless, it is reasonable to assume that a number of the here-proposed mechanisms require validation by a genetic loss-of-function approach, especially when extrapolating these results to the human system.

To this end, we set out to explore whether inflammasome components are indeed functional in human T cells. To address this, we employed a number of tool-compounds in combination with CRISPR/Cas9-based genetic perturbations. Doing so, we identified a critical role for the CARD8 inflammasome in governing pyroptosis in human primary T cells.

## Results

### Dipeptidyl-peptidase inhibition triggers a lytic cell death in primary human T cells

To elucidate whether human primary T cells are inflammasome-competent, we isolated naïve and memory CD4 and CD8 T-cell subsets from peripheral blood of healthy donors. At the same time, we also generated monocyte-derived macrophages (MDMs) from these donors. We then subjected these cells to different stimuli that are known to engage distinct inflammasome pathways in human myeloid cells: The $K^+$ ionophore Nigericin was employed to stimulate the NLRP3 inflammasome. The anthrax toxin lethal factor fused with the *Burkholderia* T3SS needle protein (YscF) was complexed with protective antigen (PA) to activate the NAIP/NLRC4 inflammasome (NeedleTox). Moreover, the DPP-inhibitor Val-boroPro (VbP) was used to trigger NLRP1 or CARD8 inflammasome activation. As an additional control, we included the combination of the BCL2-inhibitor ABT737 and the MCL1-inhibitor S63845 to engage intrinsic apoptosis. With the availability of a specific NLRP3 inhibitor (MCC950), we also included MCC950 to infer NLRP3 inflammasome activation for the Nigericin-treated conditions. In light of the fact that human monocytes require priming for IL-1β production and sufficient NLRP3 engagement, we also included a short course of LPS priming for the MDM stimulation experiments. As a measure for inflammasome activation, we assessed pyroptosis using LDH as a proxy, as well as IL-1β and IL18 release into the supernatant. Nigericin and NeedleTox treatment led to the expected outcomes in human MDMs: Nigericin triggered pyroptosis, IL-1β and IL-18 release in LPS-primed, but not in unprimed cells, while this response was fully blocked by MCC950 (Fig 1A–C). NeedleTox resulted in pyroptosis and IL-18 release in both unprimed and primed cells and again IL-1β release was only seen upon LPS priming. VbP treatment, on the other hand, led only to a small increase in LDH and IL-18 release in human monocytes and also the IL-1β response in LPS-primed cells was substantially lower compared to Nigericin or NeedleTox-treated cells. Primary T cells treated with NeedleTox showed no signs of cell death, while Nigericin treatment resulted in LDH release in both naïve CD4 and CD8 T cells.

However, unlike for MDMs, this Nigericin-dependent cell death could not be blocked by MCC950 and hence constituted an NLRP3-independent response. VbP, on the other hand, led to robust LDH release in both naïve and memory CD4 and CD8 T cells. Of note, no IL-1β or IL-18 release was observed in stimulated T cells for any of the conditions tested. Interestingly, ABT737/S63845 treatment also resulted in a substantial lytic cell death response in primary T cells, as inferred from LDH release. While the here-studied T-cell populations were highly pure, we wanted to exclude the possibility that impurities in the cell preparations impacted on the cell death-inducing activity of VbP. To this end, we subjected primary CD4 T cells to single cell cloning and tested these cells for VbP-induced cell death. These experiments confirmed that VbP triggered cell death in human T cells, while the overall responses in clonally expanded T cells were lower as compared to resting T cells (Fig EV1A and Fig 1A). In summary, these results suggested that primary T cells are highly sensitive to DPP inhibition by VbP, resulting in a lytic type of cell death that is suggestive of pyroptosis.

### VbP treatment triggers pyroptosis in T cells

To further investigate the nature of the cell death elicited by these stimuli, we performed live-cell imaging, in which we compared primary CD4 T cells and MDMs over a period of 16 h (Fig 2A and B, Fig EV2A and B, Movies EV1–EV8). We included the nuclear DNA stain propidium iodide (PI) as a marker of cell membrane integrity. Within the first hour of stimulation, the NAIP/NLRC4-stimulus NeedleTox induced rapid membrane swelling and PI uptake in MDMs, the characteristic features of pyroptosis (Movie EV1, Fig EV2B). However, in line with the LDH measurements, CD4 T cells were completely unaffected by NeedleTox treatment (Movie EV2). Treatment of macrophages with VbP induced the same characteristic morphology as NeedleTox treatment. However, the individual cells did not die simultaneously and it took up to 12 h for the last cell to commit to pyroptosis (Movie EV3, Fig EV2B). Strikingly, T cells treated with VbP also displayed characteristic features of pyroptosis, showing uniform ballooning of their cytoplasm accompanied by immediate PI uptake (Movie EV4, Fig EV2B). Similar to the VbP-treated macrophages, this followed an asynchronous—albeit faster—kinetic. As expected, treatment of T cells with ABT737/S63845 induced hallmarks of apoptosis, such as cell shrinkage and membrane blebbing (Movie EV6). At later time points, apoptotic cells also turned PI positive (Fig 2A, Fig EV2B), consistent with the concept of secondary necrosis that was also documented by LDH release (Fig 1A). To further corroborate that VbP-treated T cells undergo pyroptosis, we analyzed the cleavage of caspase-1 and its substrate GSDMD by immunoblot over a period of 48 h (Fig 2C). These experiments showed that caspase-1 was cleaved in T cells upon VbP stimulation, but not following ABT737/S63845 treatment. Moreover, VbP led to the appearance of the N-terminal 30 kDa fragment of GSDMD, which is associated with its cleavage at Asp275 and subsequent pore formation. In contrast, induction of intrinsic apoptosis readily resulted in caspase-3 maturation and PARP cleavage, as well as the formation of a 45 kDa fragment of GSDMD. The latter finding is consistent with a previous study that has shown that GSDMD can be cleaved by caspase-3 at Asp87 during apoptosis, resulting in a non-productive N-terminal (p13) and C-terminal (p45) fragment (Taabazuing *et al*, 2017). While in macrophages

NeedleTox and Nigericin induced caspase-1 and GSDMD cleavage in a priming independent and dependent manner respectively, these same treatments did not lead to the appearance of the pyroptotic GSDMD fragment in the lysate or processed caspase-1 in the supernatant of T cells (Fig EV2C). While expression data (see below)

indicate that the resistance of T cells to NeedleTox is due to the absence of NLRC4 expression, we cannot formally exclude the possibility that the toxin cannot enter T cells. Nevertheless, previous work has shown that human T cells are in principle amenable to protective antigen-mediated protein uptake (Paccani *et al*, 2005).

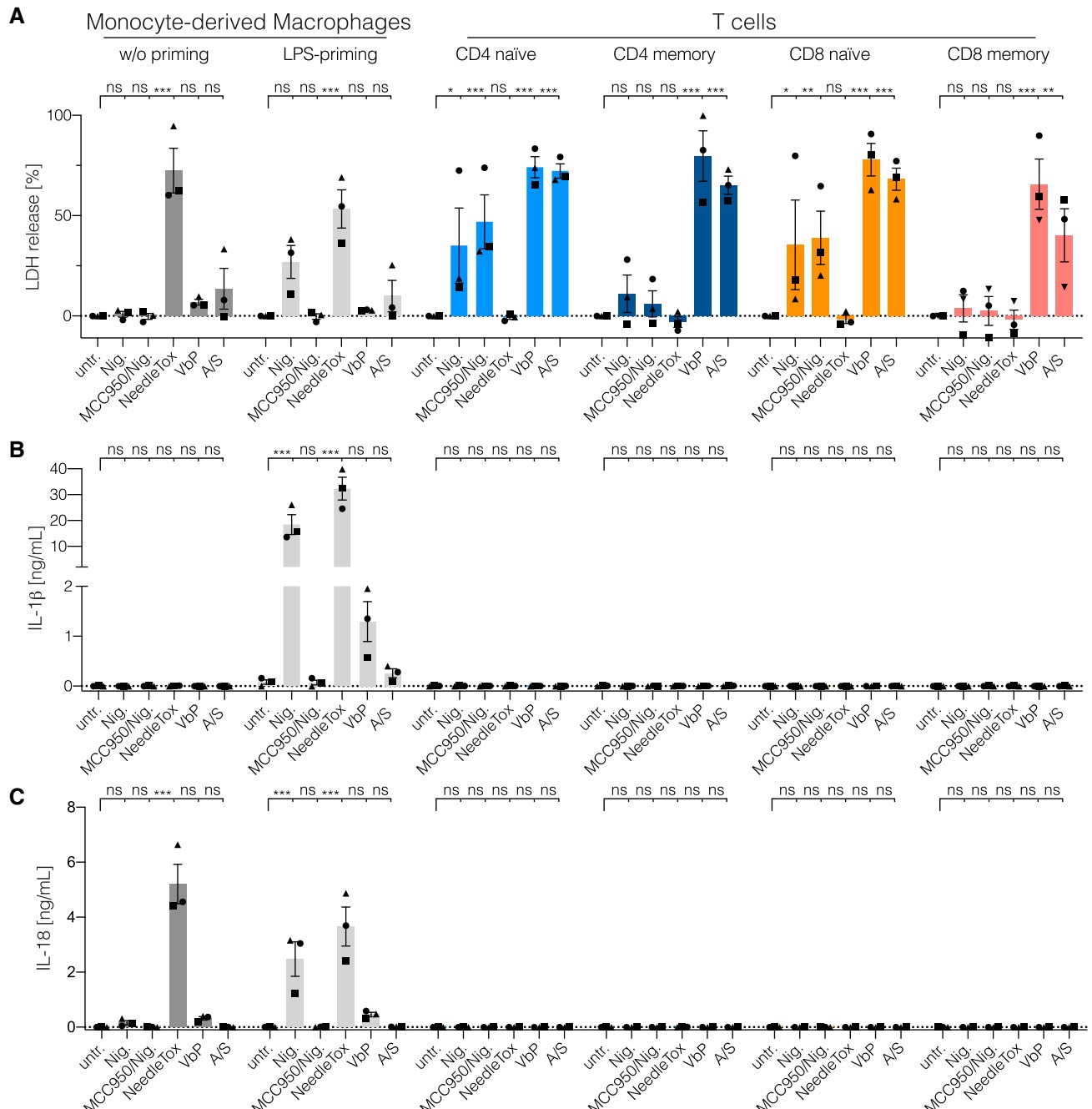

**Figure 1.  Dipeptidyl-peptidase inhibition triggers a lytic cell death in primary human T cells.**

A–C  FACS-sorted T-cell populations and Monocyte-derived Macrophages were treated with the indicated stimuli: Nigericin (Nig. 4 h), NeedleTox (4 h), Val-boroPro (VbP 22 h), and ABT737/S63845 (A/S 22 h). When indicated, cells were primed with LPS for 2 h prior to stimulation. When indicated, MCC950 was added to the media 30 min prior to the addition of Nigericin. LDH activity (A) and IL-1β and IL-18 concentration (B and C, respectively) in the supernatant were determined by LDH cytotoxicity assay and ELISA, respectively. Individual data points ± SEM from three independent donors are shown. Statistics indicate significance by two-way ANOVA: ***P ≤ 0.001; **P ≤ 0.01; *P ≤ 0.05; ns, not significant. P-values were corrected for multiple comparisons (Dunnett).

Interestingly, Nigericin treatment induced PARP cleavage in T cells, suggesting that the cell death in T cells induced by Nigericin is apoptosis with subsequent secondary necrosis rather than pyroptosis. In line with this concept, Nigericin also induced the production of the apoptotic GSDMD p45 fragment. In summary, these experiments suggested that T cells are capable of undergoing bonafide pyroptosis upon VbP treatment.

### The CARD8 inflammasome is functional in human T cells

VbP treatment is known to either activate NLRP1 (e.g. in keratinocytes) or CARD8 (e.g. in myeloid leukemia cell lines; Johnson *et al*, 2018; Zhong *et al*, 2018). Studying the myeloid cell line THP-1, we confirmed that myeloid cells engage CARD8 downstream of DPP inhibition (Fig EV3A–C). Of note, in this setting we did not observe a role for CARD8 in negatively regulating the NLRP3 inflammasome in human myeloid cells, as it has previously been suggested (Ito *et al*, 2014; Mao *et al*, 2018). Gene expression profiles from a publicly available dataset indicated that T cells express similar amounts of NLRP1 and CARD8 as monocytes, while other inflammasome sensors such as NAIP, AIM2, NLRP3, PYRIN (MEFV), or NLRC4 were scarcely expressed in T cells (Fig 3A). Immunoblotting confirmed that human T cells express both CARD8 and NLRP1, even at higher levels than MDMs (Fig 3B). GSDMD and CASP1 were expressed at comparable levels as in macrophages, while ASC expression was higher in macrophages (Fig 3B). These findings suggested that T cells employ either NLRP1, CARD8 or both sensors to respond to DPP inhibition. Indeed, consistent with the notion that NLRP1- or CARD8-inflammasome activation upon VbP treatment relies on intact proteasomal activity, cell death induction upon VbP treatment was completely blocked in T cells treated with the proteasome-inhibitor bortezomib (BTZ) in a dose-dependent manner (Fig EV2D) (Okondo *et al*, 2018). To address whether NLRP1 and/or CARD8 were required for VbP-dependent inflammasome activation in a loss-of-function setting, we generated polyclonally edited CD4 T cells deficient in *CARD8*, *NLRP1*, *PYCARD*, *CASP1*, or *GSDMD* using CRISPR/Cas9 ribonucleoprotein (RNP) complexes (Schumann *et al*, 2015). By employing two guide RNAs against each target gene, we achieved highly efficient polyclonal knockout rates as documented by sequencing of the respective target regions for InDel mutations (Fig EV4A). Studying these cells for the expression of respective target genes by immunoblot revealed efficient loss of protein expression for all of these components (Fig 3C). Next, we treated these cells with either VbP or ABT737/S63845 and then assessed LDH release as a proxy of cell death (Fig 3D). These experiments revealed that deficiency for *CARD8*, *CASP1* and *GSDMD* rendered T cells resistant to VbP-induced pyroptosis, while deficiency for *NLRP1* and *PYCARD* had no effect (Fig 3D and Fig EV4B). Secondary necrosis upon apoptosis induction was unaltered in all genotypes examined (Fig 3D). In parallel, for every genotype, we generated several clones with all-allelic frameshift mutations. We pooled these clones in order to obtain sufficient cellular material and to avoid potential clonal effects. Wild-type clones from the same donor that arose during the procedure, due to inefficient editing, were also pooled. We then treated these cells with VbP and monitored PI uptake over a period of 14 h by live-cell imaging. Here, we also observed CARD8-, caspase-1- and GSDMD-dependent, but NLRP1- and ASC-

independent pyroptosis in T cells (Fig EV4C). Altogether, these results suggested that T cells employ CARD8 to trigger inflammasome activation upon VbP treatment in an ASC-independent fashion.

### DPP9 inhibition operates upstream of CARD8 engagement

As VbP inhibits several DPPs, including DPP7, DPP8, DPP9, and DPPIV, we wished to address which is the relevant DPP responsible for preventing pyroptosis in human T cells. Previous studies have indicated that DPP8 and DPP9 are the most relevant targets for VbP-dependent inflammasome activation, with DPP9 playing a more important role (Okondo *et al*, 2017). To address whether DPP8 and DPP9 restrain CARD8 activation in primary T cells, we decided to conduct a genetic dropout experiment (Fig 4A). To this end, we targeted DPP8 or DPP9 in primary T cells using CRISPR/Cas9 in a polyclonal cell pool. Within these cell pools, we then monitored the abundance of wild-type or InDel mutations at the respective target loci over time using deep sequencing. Doing so, we observed a steady increase in wild-type reads for the DPP9-, but not DPP8-targeted cell population, indicating that loss of DPP9 but not DPP8 is lethal for the cells. Of note, this increase in wild-type reads was not observed when DPP9 was concomitantly targeted with CARD8, thus indicating an epistatic connection of DPP9 to CARD8 (Fig 4B and C). We therefore conclude that DPP9 but not DPP8 is the relevant DPP that restricts the activation of the CARD8 inflammasome and execution of pyroptosis in human T cells.

### TCR stimulation renders T cells resistant to pyroptosis

In the preceding experiments, T cells could tolerate DPP9 deficiency for several days without overt signs of pyroptosis, while it took up to 14 days for the selective pressure on DPP9 deficiency to be fully reflected in sequencing data. Furthermore, when generating knockout T cells, we noted that resting T cells following expansion increased the susceptibility to VbP-induced pyroptosis. In both cases, the experimental setup entailed the use of anti-CD3, anti-CD28 beads (CD3/CD28 beads) that resulted in the polyclonal activation of T cells. To test the impact of T-cell activation on CARD8 inflammasome engagement, we compared freshly isolated primary CD4 T cells that were either activated with CD3/CD28 beads for 4 days or left untreated. Doing so, we observed a marked decrease in susceptibility to subsequent stimulation with VbP (Fig 5A). In contrast, secondary necrosis upon ABT737/S63845 treatment was only marginally reduced. Interestingly, upon T-cell receptor (TCR) stimulation, we noted an upregulation of full-length CARD8 that was accompanied by a slight decrease of the cleaved C-terminal fragment. As autocleavage is a prerequisite for CARD8-activation, the alteration in the pool of autocleaved and therefore activatable CARD8 could indicate a potential switch which regulates this pathway in T cells. Other components of the CARD8 inflammasome cascade (CASP1 and GSDMD) showed—despite great inter-donor variability—no unambiguous alteration in expression levels (Fig 5B and C). Altogether, these results suggest that T-cell activation either results in the upregulation of proteins that inhibit the CARD8 inflammasome pathway or that inhibitory post-translational modifications on key components such as on CARD8 itself are induced.

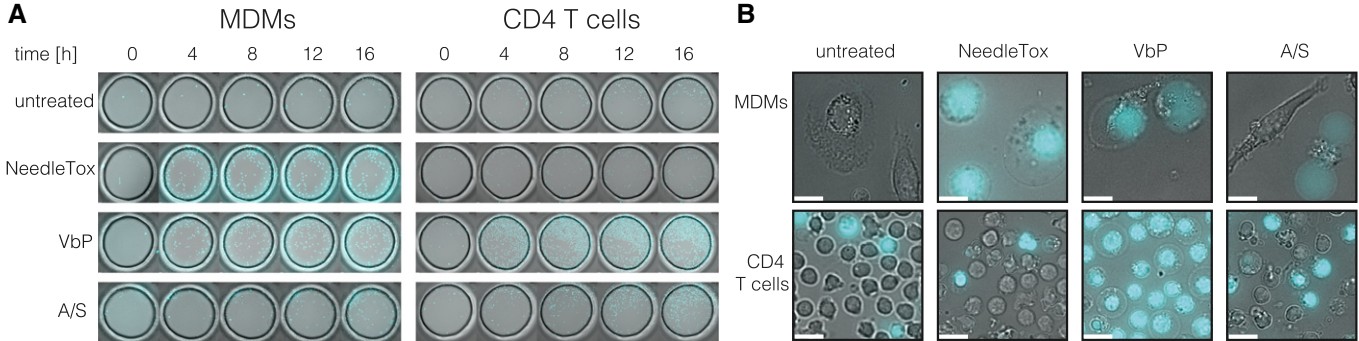

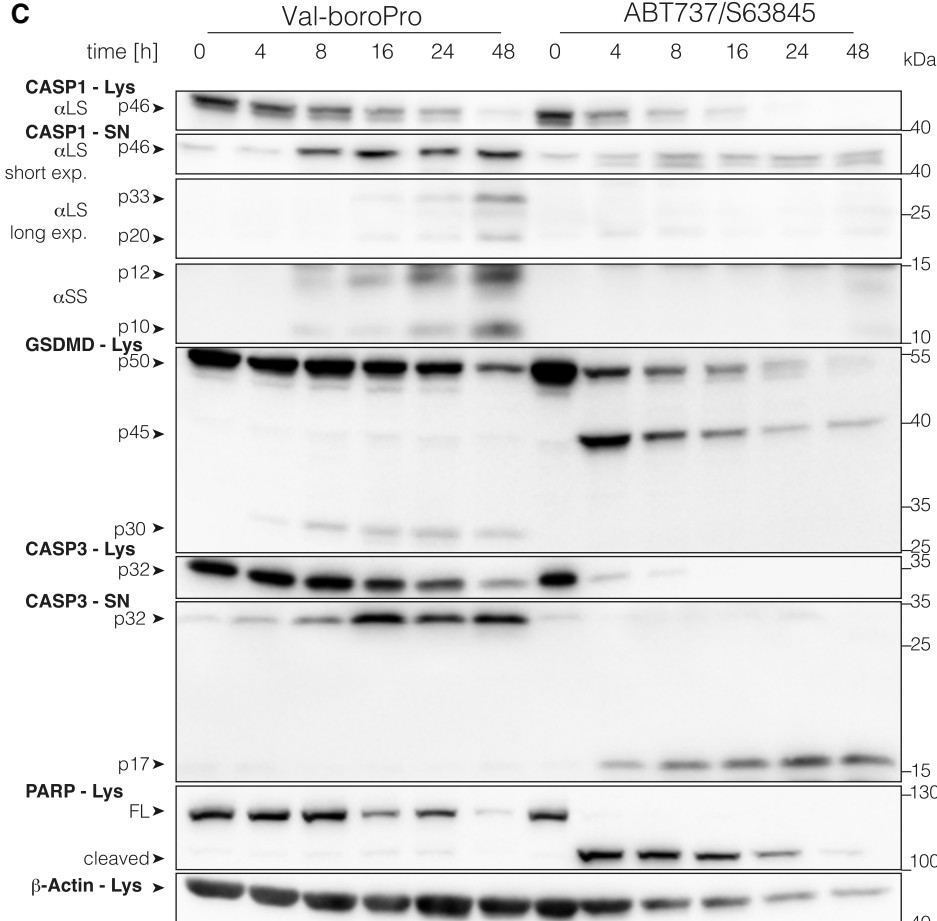

**Figure 2. VbP-treated T cells display hallmarks of pyroptosis.**

A   Macrophages and CD4 T cells from the same donor were subjected to the indicated treatments and morphologic changes as well as PI uptake were monitored by live-cell imaging microscopy using a 25× objective. Representative images from indicated time points are shown. Cyan color coding is used for the fluorescent PI signal. One donor out of two is shown.

B   Representative images were acquired with a 63× objective at the end of the experiment shown in (A) at 16 h. One donor out of two is shown. Scale bars: 25 μm.

C   CD4 T cells were treated with VbP or ABT737/S63845. Lysate and supernatant samples were collected at the indicated time points. Samples were analyzed by immunoblotting. αSS and αLS indicate the use of a small subunit or large subunit-specific caspase-1 antibody respectively. Lys = lysate, SN = supernatant, FL = full length. One representative experiment out of three is shown.

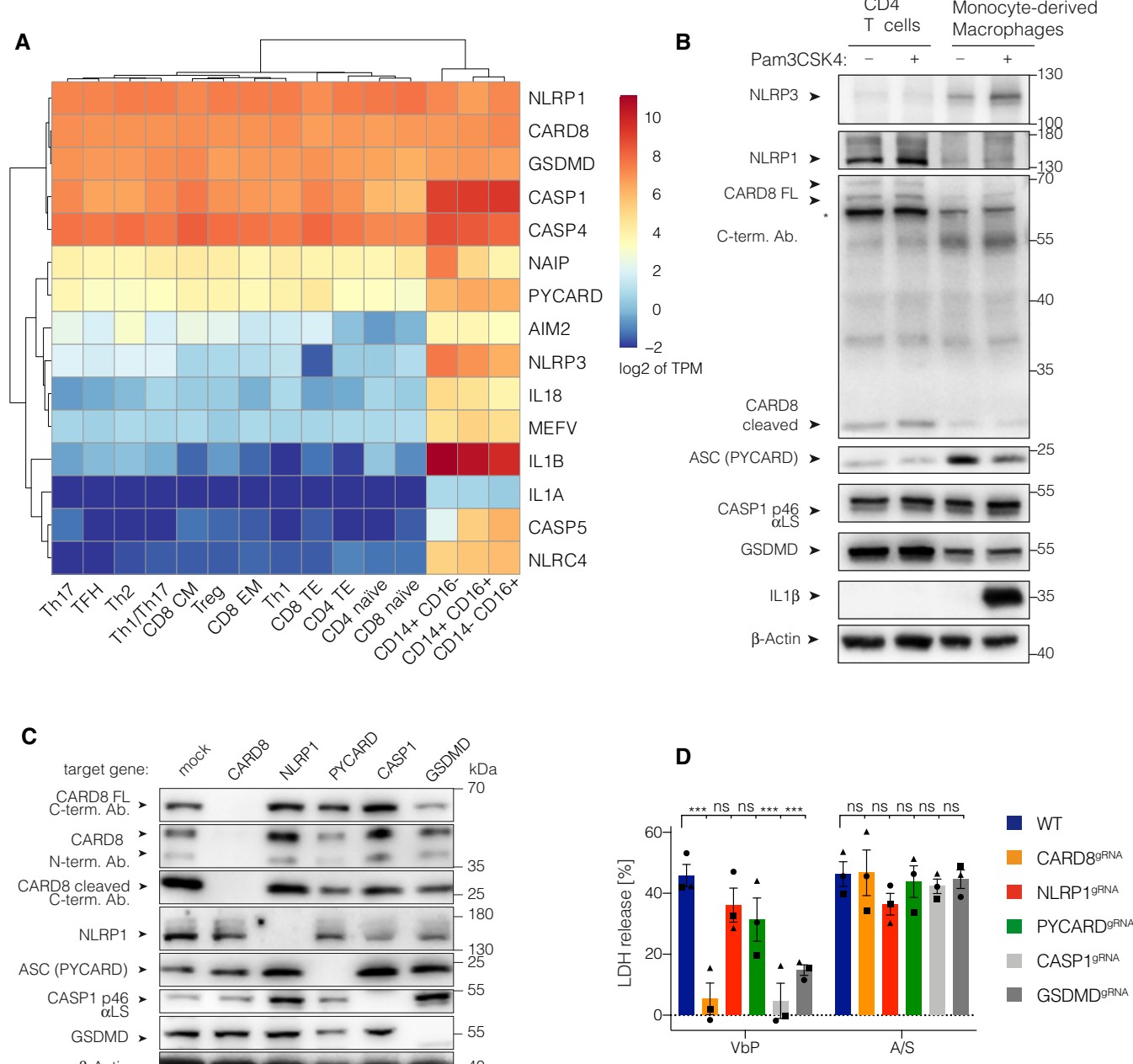

**Figure 3. The CARD8 inflammasome is functional in human T cells.**

A   Gene expression profile of 15 inflammasome-related genes in human T cell and monocyte subsets. TPM values were retrieved from a public RNASeq dataset and log2 transformed. Both, rows and columns are clustered using euclidean distance and complete linkage. Tfh, T follicular helper cells; Tregs, T regulatory cells; Th, T helper cells; CM, central memory T cells; EM, effector memory T cells; TE, terminal effector T cells; CD14$^+$C16$^-$, classical monocytes; CD14$^+$CD16$^+$, intermediate monocytes; CD14$^-$ CD16$^+$, non-classical monocytes.

B   Immunoblotting of CD4 T cells and MDMs treated with Pam3CSK4 for 6 h. * indicates an unspecific band. One representative experiment out of three is shown.

C   Immunoblotting of polyclonal CD4 T cells targeted with two gRNAs for each indicated target gene. One representative experiment out of three is shown.

D   CD4 T cells from the indicated polyclonal gene targeting approach were treated with the indicated stimuli for 18 h. Individual data points ± SEM from three independent donors are shown. Statistics indicate significance by two-way ANOVA: ***$P \leq 0.001$; ns, not significant. *P*-values were corrected for multiple comparisons (Dunnett).

## Discussion

Here, we show that human primary T cells are fully competent for conventional inflammasome signaling, governing the execution of

pyroptosis. Testing different stimuli that are known for their inflammasome-agonistic activity in myeloid cells, we uncovered a unique role for the DPP-inhibitor VbP to trigger inflammasome activation in T cells. Upon VbP treatment, we observed that T cells

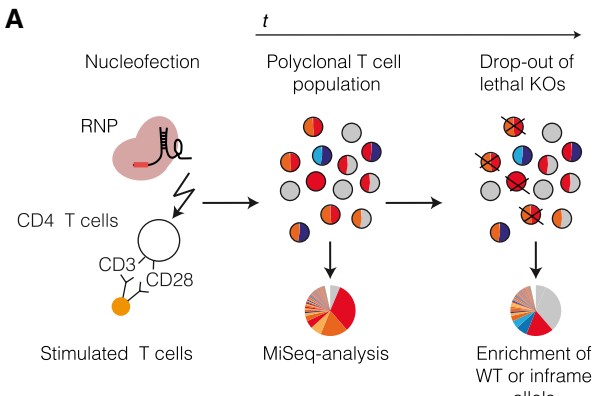

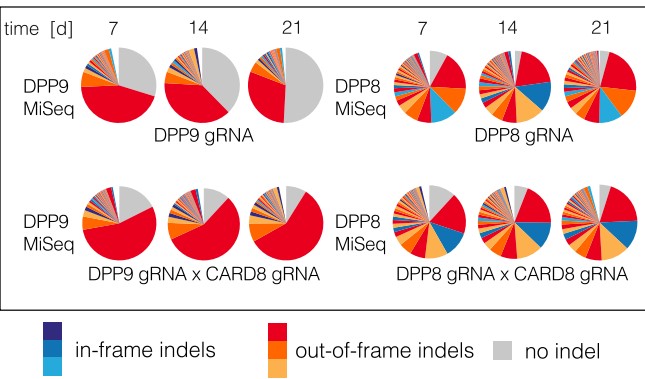

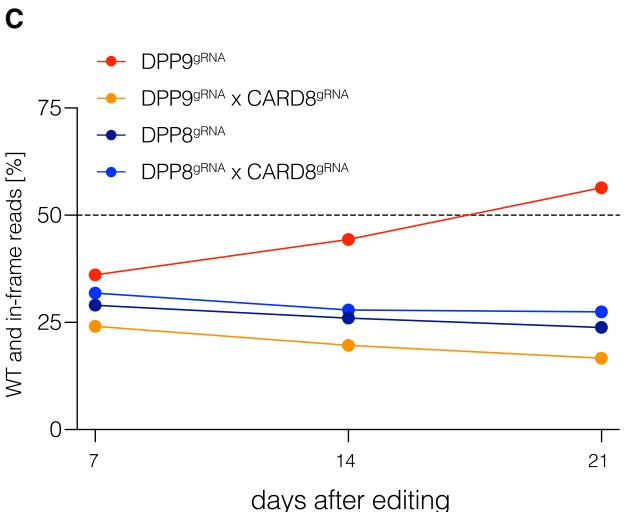

**Figure 4. DPP9 inhibition operates upstream of CARD8 engagement.**

A Schematic of the experimental setup.

B MiSeq analysis of the indicated gene targeting approaches from samples collected at the indicated time points post-editing. DPP9 and DPP8 were targeted with one gRNA/target gene; CARD8 was targeted with two gRNAs. On the day of nucleofection, T cells were activated with human T activator CD3/CD28 beads and expanded in the presence of IL-2. On day 10, cells were switched to IL-7/IL-15 containing medium.

C Fraction of wild-type and in-frame reads is depicted. One representative experiment out of three is shown.

succumbed to a lytic form of cell death that affected the entire cell population, yet with delayed kinetics. Interestingly, no major difference in susceptibility to VbP was observed when studying CD4 and CD8 T cells of naïve or memory phenotype. Assessing the phenotype of VbP-induced cell death in T cells revealed characteristic features of pyroptosis, and immunoblotting indicated that both caspase-1 and GSDMD were cleaved in these cells. In light of previous work on the role of DPP-inhibitors in the context of inflammasome activation, we focused on the role of NLRP1 or CARD8 in these cells. Interestingly, while both these sensors were expressed at considerable amounts in T cells, it turned out that CARD8, but not NLRP1, was required to trigger VbP-induced pyroptosis. As such, CARD8-deficient T cells were fully protected from VbP-induced pyroptosis. In line with the notion that CARD8 can directly engage caspase-1, CARD8-dependent pyroptosis proceeded independently of ASC (Ball et al, 2020). Further genetic dissection revealed that CARD8-induced pyroptosis required caspase-1 as well as GSDMD, although a small proportion of GSDMD-independent cell death was observed. Intriguingly, CARD8-triggered pyroptosis could be largely reverted by the engagement of T-cell receptor (TCR) signaling. TCR engagement showed the trend of a relative decrease of the cleaved CARD8 version in relation to the full-length protein. As autocleavage has been shown to be an essential prerequisite for the activatability of CARD8, this could point toward a mechanistic explanation for this phenomenon. Overall, it appears most likely that a post-translational mechanism downstream of TCR signaling impedes inflammasome activation in T cells. At the same time, it is conceivable that TCR-induced de novo expressed proteins negatively affect CARD8 activation. Finally, studying the role of DPP8 and DPP9 upstream of CARD8 inflammasome activation using a genetic dropout approach, we found that DPP9, but not DPP8 activity, was critically required to restrain CARD8 activation. In summary, these results suggest that resting human T cells are inflammasome-competent cells that employ the CARD8-CASP1-GSDMD signaling axis to engage in a lytic type of cell death. Under steady state conditions, this pathway is negatively regulated by the catalytic activity of DPP9 upstream of CARD8.

While previous work has well established that myeloid cells and myeloid cell lines are sensitive to VbP-induced pyroptosis downstream of CARD8, lymphocytes were considered to be insensitive to this pathway. As such, T-cell levels in VbP-treated mice remained unaffected, and Jurkat T cells as well as primary T-ALL cells were found to be VbP-resistant, which was ascribed to low expression of CASP1 and CARD8 in these cells (Johnson et al, 2018). Studying human primary cells, we observed that VbP is indeed more potent in inducing pyroptosis in primary T cells than in myeloid cells. Interestingly, the cell death-inducing capacity of VbP in T cells has already been reported in 1999 (Chiravuri et al, 1999). Here, it was shown that VbP-induced cell death in resting, but not proliferating T cells, and it was furthermore reported that proteasome inhibitors block this response. These phenomena were linked to the inhibition of DPP7, and the downstream cell death was ascribed to be apoptosis. With our study, we can clearly show that the cell death downstream of VbP is CARD8-dependent pyroptosis rather than apoptosis and that the involved DPP is DPP9 rather than DPP7 (Chiravuri et al, 1999). The potent cell death induced by VbP in human T cells raises concerns regarding the safety of this compound that is currently being evaluated as an anti-cancer agent in several clinical

**A**

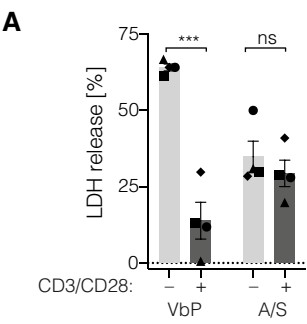

Figure 5. TCR stimulation renders T cells resistant to pyroptosis.

A CD4 T cells were activated with human T activator CD3/CD28 beads (bead:cell ratio 1:5) for 4 days. Controls were kept resting in the presence of IL-7 and IL-15. On day 4, cells were stimulated as indicated for 18 h and cytotoxicity was assessed by LDH-activity in the supernatant. Individual data points ± SEM from four independent donors. Statistics indicate significance by two-way ANOVA: ***$P \leq 0.001$; ns, not significant. $P$-values were corrected for multiple comparisons (Sidak).

B Immunoblotting of CD4 T cell lysates treated as in (A) for 3 to 4 days. * indicates an unspecific band.

C Quantification of immunoblots shown in (B) and additional donors ($n = 9$ for all but GSDMD, GSDMD $n = 8$, for one donor quantification of the GSDMD band was considered unreliable due to uneven substrate distribution). Individual data points ± SEM are shown. Statistics indicate significance by one-way ANOVA of log2-transformed fold change values of band intensities (activated/non-activated T cells) normalized to β-Actin: ***$P \leq 0.001$; *$P \leq 0.05$; ns, not significant. $P$-values were corrected for multiple comparisons (Sidak). All band intensity ratios were tested against β-Actin. CARD8 full length was additionally tested against CARD8 cleaved.

**B**

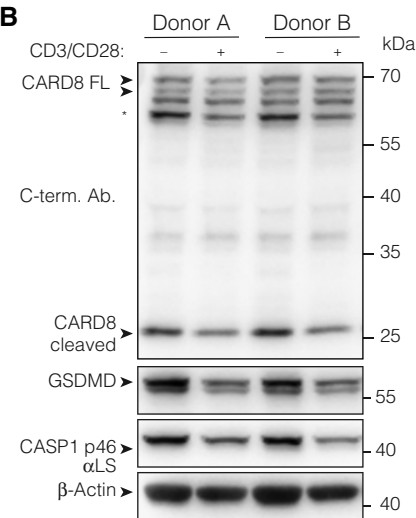

**C**

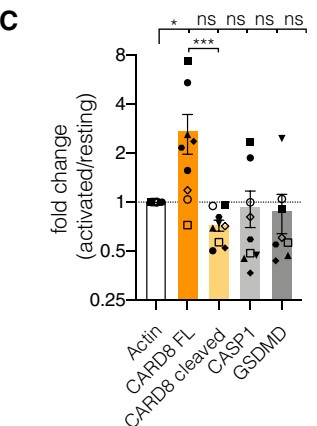

et al, 2015; Arbore et al, 2016; Martin et al, 2016), we could not observe NLRP3-inflammasome functionality in these cells. However, it has to be noted that we only applied Nigericin to test the activity of this pathway in T cells. Nigericin is a well-established and selective agonist for the NLRP3 inflammasome by inducing $K^+$ efflux, yet other $K^+$ efflux-independent pathways have been described to engage the NLRP3 inflammasome as well. As such, we cannot fully exclude that NLRP3 can indeed be engaged to induce inflammasome activation in T cells. However, in line with its low expression, we speculate that NLRP3 would only play a subordinate if any role at all. At the same time, our current studies cannot exclude the possibility that NLRP3 or other inflammasome components play a role in determining T-cell fate, as it has been proposed for murine Th2 or Th17 differentiation. Nevertheless, with the availability of genome engineering tools that allow the manipulation of naïve, primary T cells, these hypotheses can be addressed employing appropriate ex vivo models.

The high sensitivity of T cells to CARD8 inflammasome engagement obviously raises the question of the physiological role of this pathway. Lytic, programmed cell death pathways such as pyroptosis are thought to have evolved to sense intracellular microbial infection. In this context, cell death constitutes an "altruistic" behavior of the infected cell to avoid further microbial replication. Moreover, its execution in a lytic form as opposed to apoptosis serves to release cytosolic content, which can further amplify antimicrobial immunity by alarming bystander cells (see below). In this regard, it is conceivable that the DPP9-CARD8-CASP1-GSDMD axis functions as a sensor for microbial infection. As such, T-cell tropic viruses would pose the most likely candidates for being sensed by this pathway. Indeed, the fact that T cells are pyroptosis-competent has already been proposed in the context of HIV infection. Here, it was shown that abortive HIV infection of quiescent T cells resulted in caspase-1-dependent pyroptosis (Doitsh et al, 2010, 2014). The HIV-derived substrates triggering this response were found to be incomplete reverse transcripts that accumulate in abortively infected cells. Later studies of the same group ascribed this infection to be sensed by IFI16, which contains an N-terminal pyrin domain and a C-terminal nucleic acid binding domain (HIN domain; Monroe et al, 2014). Since IFI16 has been shown to employ ASC as an adapter protein to

trials (Eager et al, 2009). Our data would suggest that VbP treatment especially affects the viability of resting T cells, a phenomenon that would be underestimated if not overlooked when studying the murine system.

While several publications propose a T cell-intrinsic role of NLRP3 upstream of caspase-1 activity (Doitsh et al, 2014; Bruchard

activate caspase-1 (Singh *et al*, 2013), these studies would imply IFI16 as a direct sensor of HIV-derived nucleic acids that results in T-cell pyroptosis. However, it is also possible that the impact of IFI16 on HIV-induced pyroptosis is not due to it functioning as a direct sensor, but rather due to it impacting on another mechanism that indirectly affects the pyroptotic response of T cells. Indeed, IFI16 has been shown to impede HIV-1 propagation functioning as a restriction factor by preventing Sp1 from binding to its transcription factor-binding regions within the HIV-1 LTR (Hotter *et al*, 2019). However, apart from implying IFI16, it should be noted that the here-unveiled mechanism differs from these previous reports in two aspects. On the one hand, we did not observe IL-1β release in the context of pyroptosis, which was well documented in the HIV-infection experiments. On the other hand, HIV-induced pyroptosis was only operational in T cells isolated from tonsillar tissue, but not from peripheral blood (Munoz-Arias *et al*, 2015). Nevertheless, in light of the prominent cell death-inducing activity of the CARD8 pathway it should be interesting to revisit the role of pyroptosis in response to HIV infection or other T-cell tropic viruses.

Another, not mutually exclusive possibility is that CARD8 plays an important role in controlling T-cell selection or homeostasis. While apoptosis is the default cell death program in T-cell selection in the thymus as well as in maintenance of homeostatic levels of T cells in the periphery, it is conceivable that CARD8-driven pyroptosis also plays a role in these processes. As such, the fact that malignantly transformed T cells become resistant to VbP-driven pyroptosis suggests that a CARD8-/caspase-1-dependent cell death occurs under physiological conditions in healthy T cells and that T cells committed to malignant transformation might profit from shutting down this programmed cell death pathway. Comparing naïve and memory T cells did not reveal an association between VbP sensitivity and the differentiation state. However, our functional studies show that quiescent, but not activated, T cells are especially sensitive to this pathway. Therefore, in light of these findings it would appear most constructive to investigate the role of CARD8 in an in vivo setting, in which the dynamics of an adaptive immune response can be studied. This is obviously hampered by the absence of a murine orthologue of CARD8, which would require the use of a humanized mouse model to address this question.

The term pyroptosis was initially coined to describe a pro-inflammatory, programmed cell death pathway that was observed in macrophages following *Salmonella* infection (Cookson & Brennan, 2001). Mechanistically, this cell death could be differentiated from other cell death pathways by its dependency on caspase-1. Moreover, unlike apoptosis, this cell death resulted in rapid cell lysis and was typically associated with the maturation and release of IL-1β. The discovery of inflammasomes laid the conceptual groundwork for how the activity of caspase-1 is specifically regulated in the context of different types of cellular perturbations (Schroder & Tschopp, 2010). This caspase-1-centric definition of pyroptosis was then extended by the notion that caspase-4 could also trigger pyroptosis upon LPS recognition (Shi *et al*, 2014). Subsequently, the discovery of GSDMD as the substrate for caspase-1/4-dependent cell death induction finally shifted the definition of pyroptosis to its cell death-inducing substrate (Kayagaki *et al*, 2015; Shi *et al*, 2015). As such, it turned out that additional gasdermin molecules exist that can execute lytic cell death by pore formation independently of caspase-1/4 (Broz *et al*, 2020). At this point, with the characterization of GSDME and GSDMB, two additional gasdermin molecules are know that can execute pyroptosis upon cleavage (Rogers *et al*, 2017; Wang *et al*, 2017; Zhou *et al*, 2020). GSDME is processed by caspase-3, which results in the formation of a pore-forming fragment that functions analogous to the one of GSDMD. These studies showed that the phenotype of caspase-3-triggered cell death depends on the availability of its potential substrate GSDME, which—if present—switches apoptosis to pyroptosis. Despite the fact that IL-1β is neither present in GSDME-expressing cells nor matured by caspase-3, GSDME-dependent pyroptosis downstream of caspase-3 turned out to be inflammatory (Wang *et al*, 2017; Liu *et al*, 2020). Here, it was suggested that cytosolic content, released in the context of pyroptosis, results in the activation of bystander cells. While fulfilling the genetic definition of pyroptosis by being a GSDMD-dependent cell death, we have thus far been unable to identify the release of pro-inflammatory mediators from T cells succumbing to CARD8 activation. As such, we were unable to detect the release of cytokines of the IL-1 family in supernatants from VbP-treated T cells. While these observations do not yet rule out that GSDMD triggered cell death in T cells is pro-inflammatory and that T cells can acquire expression of IL-1-related cytokines under certain pathogenic circumstances, these current data would indeed imply a complete disconnect of "pyroptosis" from its initial definition.

# Materials and Methods

### Reagents and Tools table

| Reagent/Resource | Reference or Source | Identifier or Catalog Number |
|---|---|---|
| **Experimental Models** | | |
| Peripheral blood mononuclear cells (PBMCs) of healthy donors from leucoreduction system chambers | Department of Transfusion Medicine, University Hospital, LMU Munich | |
| THP-1 cells | ATCC | TIB-202 |
| **Antibodies** | | |
| CD3-PerCP, clone UCHT1, mouse IgG1, 1:100 | BioLegend | #300428 |
| CD4-Qdot605, clone S3.5, mouse IgG2a, 1:200 | ThermoFisher | #Q10008 |

**Reagent and Tools table** (continued)

| Reagent/Resource | Reference or Source | Identifier or Catalog Number |
|---|---|---|
| CD8-BV510, clone SK1, mouse IgG1, 1:200 | BD | #563919 |
| CD45RA-PE, clone HI100, mouse IgG2b, 1:200 | BioLegend | #304108 |
| CCR7-BV421, G043H7, mouse IgG2a, 1:25 | BioLegend | #300428 |
| Human GSDMDC1 Ab., rabbit, polyclonal, 1:1000 | Novus Biologicals | #NBP2-33422 |
| Human Caspase-1 (p20), mAb (Bally-1) (=αLS Ab.), 1:1000 | Adipogen International | #AG-20B-0048-C100 |
| Human CASP1 p46 p12 p10 antibody, rabbit (=αSS Ab.), monoclonal, 1:1,000 | Abcam | #ab179515 |
| Human PARP-Ab., 1:1,000 | Cell Signalling Technology | #9542 |
| Human CASP3-Ab., 1:1,000 | Cell Signalling Technology | #9962 |
| Human β-Actin-Ab., monoclonal, HRP-conjugated, 1:1,000 | Santa Cruz | #sc-47778 |
| Human CARD8 c-terminal Ab., mouse, polyclonal, 1:1,000 | Abcam | #ab24186 |
| Human CARD8 N-terminal Ab., rabbit, polyclonal, 1:1,000 | Abcam | #ab194585 |
| Human NLRP1 Ab., mouse, monoclonal, 1:1,000 | BioLegend | #679802 |
| Human ASC Ab., rabbit, polyclonal, 1:1,000 | Adipogen | #AG-25B-0006-C100 |
| Human IL-1 beta /IL-1F2 Antibody, goat, polyclonal, 1:1,000 | R&D Systems | #AF-201-NA |
| Anti-rabbit IgG, HRP-linked Antibody, 1:1,000 | Cell Signalling Technology | #sc-47778 |
| Donkey anti-goat IgG-HRP, 1:1,000 | Santa Cruz | #sc-2020 |
| Anti-mouse IgG, HRP-linked Antibody, 1:1,000 | Cell Signalling Technology | #7076P2 |
| Human NLRP3 Ab. (Cryo-2), monoclonal, 1:1,000 | Adipogen | #AG-20B-0014-C100 |
| **Oligonucleotides and sequence-based reagents** | | |
| gRNAs | This study, synthesized by IDT and Synthego | Table EV1 |
| **Chemicals, enzymes and other reagents** | | |
| Human IL-1β ELISA Set II | BD Biosciences | #557953 |
| Propidium iodide | eBioscience | #00-6990-50 |
| RBC Lysis Buffer (10X) | Biolegend | #420301 |
| Biocoll density 1.077 g/ml, isotone | Merck Milipore | #L6115 |
| CD14 microbeads, human | Miltenyi Biotec | #130-050-201 |
| Pan T cell isolation kit human | Miltenyi Biotec | #130-096-535 |
| Naive CD4 T cell isolation kit II, human | Miltenyi Biotec | #130-094-131 |
| CD4 T cell isolation kit, human | Miltenyi Biotec | #130-096-533 |
| LS columns | Miltenyi Biotec | #130-042-401 |
| FulTrac Micro inserts | IBIDI | #80489 |
| μ-Plate 24-well black | IBIDI | #82406 |
| Human Serum from AB, male clotted whole blood | Sigma | #H5667 |
| Penicillin-Streptomycin (10,000 U/ml) | Thermo Fisher Scientific | #15140163 |
| RPMI 1640 Medium, no phenol red | Thermo Fisher Scientific | #11835063 |
| RPMI 1640 Medium | Thermo Fisher Scientific | #21875-034 |
| Hepes solution, 1 M, pH 7.0-7.6, sterile-filtered, BioReagent | Sigma-Aldrich | #H0887-100ML |
| Pyruvate | Gibco | #11360070 |
| Recombinant Human IL-7 | PeproTech | #200-07-50ug |
| Recombinant Human IL-15 | PeproTech | #200-15-50ug |
| Recombinant human IL-2 | R&D Systems | #202-IL-500 |
| LPS-EB Ultrapure | InvivoGen | #tlrl-3pelps |

**Reagent and Tools table** (continued)

| Reagent/Resource | Reference or Source | Identifier or Catalog Number |
|---|---|---|
| Recombinant Cas9 protein from *S. pyogenes* | this study, Max-Planck-Insitute of Biochemistry, Munich | |
| Bortezomib | Selleck Chemicals | #S1013 |
| ABT737 | Selleck Chemicals | #S1002 |
| S63845 | Selleck Chemicals | #S8383 |
| Nigericin sodium salt | Sigma-Aldrich | #N7143 |
| Protective antigen (pA) | Biotrend | #LL-171E |
| MCC950 (CRID3) | Tocris Biosciences | #5479 |
| Val-boroPro | APExBIO | #B3941 |
| CyQUANT™ LDH Cytotoxicity Assay | Invitrogen | #C20301 |
| LFn-YcsF | this study | |
| Brilliant Stain Buffer | BD Biosciences | #563794 |
| LIVE/DEAD Fixable Near-IR Dead Cell Stain Kit | Thermo Fisher Scientific | #L10119 |
| Pierce BCA Protein Assay Kit | Thermo Fisher Scientific | #23227 |
| cOmplete™ Protease Inhibitor Cocktail | Roche / Sigma-Aldrich | #11697498001 |
| Amersham Protran 0.2 Nitrocellulose Western Blot Membrane | GE Healthcare | #10600001 |
| Bovine Serum Albumin | Sigma-Aldrich | #A7906-100G |
| Fetal Bovine Serum (FCS) | Thermo Fisher Scientific | #10270106 |
| Phorbol 12-myristate 13-acetate (PMA) | Enzo Life Sciences | #BML-PE160-0005 |
| Dynabeads™ Human T-Activator CD3/CD28 | Gibco | #11132D |
| Pam3CSK4 | InvivoGen | tlrl-pms |
| **Software** | | |
| Outknocker 2.0 | http://www.outknocker.org/outknocker2.htm, Schmid-Burgk *et al*, 2014 | |
| Prism 8 | GraphPad | |
| Adobe Illustrator | Adobe | |
| Adobe Photoshop | Adobe | |
| **Other** | | |
| MiSeq Reagent Kit v2, 300 Cycles | Illumina | #MS-102-2002 |
| P3 Primary Cell 96-well Kit | Lonza | #V4SP-3096 |

### Isolation of PBMCs, primary human monocytes and T cells

Peripheral blood mononuclear cells (PBMCs) were isolated from residual heparinized blood retained in leukocyte reduction system chambers of healthy thrombocyte donors by density gradient centrifugation (Biocoll, Merck Millipore) and erythrocyte lysis (RBC lysis buffer, BioLegend). Informed consent was obtained from all subjects according to the Declaration of Helsinki and approval by the responsible ethical committee (project number 19-238, Ethics committee of the medical faculty of the Ludwig Maximilian University Munich). Human monocytes were MACS-purified from PBMCs using CD14 microbeads (Miltenyi). Monocytes were differentiated into monocyte-derived macrophages (MDMs) for 5–7 days in the presence of CSF-1 (200 ng/ml, Max Planck Protein Facility, Munich) with fresh CSF-1 being added on day 3 and day 5. T cells were isolated either directly from PBMCs or the CD14-depleted flow-through of the monocyte-isolation by using CD4 T-cell isolation kit or Naïve CD4 isolation kit II (all Miltenyi). For FACS-purification of T-cell subsets, T cells were pre-enriched from PBMCs using Pan T cell isolation kit (Miltenyi). T cells were cultivated in RPMI 1640 (Gibco) supplemented with 2.5% (v/v) human serum (Sigma-Aldrich), Penicillin-Streptomycin (100 IU/ml, Thermo Fisher Scientific), Pyruvate (1 mM, Gibco), and HEPES (10 mM, Sigma-Aldrich). Whenever T cells were rested for more than one night before the experiment was started, IL-7 and IL-15 (5 ng/ml each) were added to the cell culture medium.

## FACS

T cells were allowed to rest for up to three days in RPMI (2.5% human serum) in the presence of IL-7 and IL-15 (5 ng/ml each, PeproTech). Dead cells were stained with the LIVE/DEAD Fixable Near-IR Dead Cell Stain Kit (Thermo Fisher Scientific) for 20 min at room temperature in the dark. Consecutively, cells were stained with the following antibodies in brilliant stain buffer: CD3-PerCP, CD4-Qdot605, CD8-BV510, CD45RA-PE, and CCR7-BV421. Before sorting, cells were washed and resuspended in FACS buffer (PBS with 2 mM EDTA and 2% FCS). Cells were collected in FACS tubes, containing RPMI with 2.5% (v/v) human serum. Based on surface marker staining, viable cells (Near-IR neg.) were sorted on a BD FACS Aria-Fusion into CD4 naïve (CD3$^+$CD4$^+$CD8$^-$ CD45RA$^+$ CCR7$^+$), CD4 memory (CD3$^+$CD4$^+$CD8$^-$ CD45RA$^+$/CCR7$^-$, CD45RA$^-$/CCR7$^+$, CD45RA$^-$/CCR7$^-$), CD8 naïve (CD3$^+$CD4$^-$ CD8$^+$CD45RA$^+$ CCR7$^+$), and CD8 memory (CD3$^+$CD4$^-$ CD8$^+$CD45RA$^+$/CCR7$^-$, CD45RA$^-$/CCR7$^+$, CD45RA$^-$/CCR7$^-$).

## Cell stimulation

If not otherwise indicated, T cells and MDMs were plated at a concentration of 666.666 cells/ml either in 96-well (150 µl) or 384-well plates (48 µl). Stimuli and inhibitors were used at the following concentrations: Nigericin (6.5 µM, Sigma-Aldrich), MCC950 (10 µM, Tocris Biosciences), Val-boroPro (4 µM, APExBIO), ABT737 (1 µM, Selleck Chemicals), S63845 (1 µM, Selleck Chemicals), LPS (200 ng/ml), Pam3CSK4 (2 µg/ml), and Bortezomib (as indicated, 0.1–10 µM, Selleck Chemicals). NeedleTox is composed of LFn-YcSF (25 ng/ml) and protective antigen (250 ng/ml). The following cytokines were used at the indicated concentrations: IL-7 (5 ng/ml, Peprotech), IL-15 (5 ng/ml, Peprotech), and IL-2 (50 IU/ml, R&D Systems).

## THP-1 cells

THP-1 cells were cultivated in RPMI 1640 (Gibco) supplemented with 10% (v/v) FCS (Gibco), Penicillin-Streptomycin (100 IU/ml, Thermo Fisher Scientific), and Pyruvate (1 mM, Gibco). For differentiation, THP-1 cells were plated in 10-cm cell culture dishes at $1 \times 10^6$ per ml in the presence of PMA (100 ng/ml, Enzo Life Sciences) and incubated for 16 h. On the next day, non-adherent cells were removed; adherent cells were washed off the plate with PBS, resuspended in medium, and plated at 750,000 cells per ml in 96-well plates (100 µl final volume). Cells were allowed to settle down for several hours before an experiment was started.

## LDH assay

In experiments in which LDH activity was determined in the supernatant, cells were plated and stimulated in RPMI 1640 without phenol red (Gibco). LDH assay was performed according to the manufacturer's instructions (Invitrogen). Relative LDH release was calculated as LDH release [%] = 100 × (measurement − unstimulated control)/(lysis control − unstimulated control).

## Immunoblotting

For detection of proteins in the supernatant, T cells were plated a concentration of $6 \times 10^6$ cells/ml and MDMs at $2 \times 10^6$ cells/ml in RPMI containing 1.25% human serum in 600 µl in a 24-well plate. Supernatants were precipitated with methanol/chloroform in a ratio of 1:1:0.25, resuspended in 1× Laemmli sample buffer, and denatured at 95°C for 10 min (Jakobs *et al*, 2013). Cells were washed 1× in PBS and lysed in RIPA lysis buffer containing protease inhibitor (Roche), spun clear at 18,000 *g* for 10 min at 4°C and lysates were transferred to new tubes. Protein concentration was determined by BCA assay (Pierce BCA protein assay kit, Thermo Fisher Scientific) according to the manufacturer's instructions and protein amounts were adjusted between lanes. Supernatant and lysate samples were separated by tris-glycine denaturing SDS-PAGE (15% for supernatant, 12% for lysate). Proteins were blotted onto 0.2 mM nitrocellulose membranes (GE Healthcare), blocked in 3% BSA (Sigma-Aldrich) in TBS-T, and incubated with the indicated primary (in 3% BSA TSBT) and corresponding secondary antibodies (in 3% milk). Chemiluminescent signals were recorded on a Fusion Fx (Vilber) with a CCD-camera, and respective images were contrast-enhanced in a linear fashion. The uncropped images of the immunoblots are available as Figure Source Data.

## Quantification of immunoblots

Band intensities from non-saturated immunoblots were measured with the built-in Gel Analyzer tool of the ImageJ software. Intensities of the protein of interest were normalized to corresponding β-Actin bands of the same lane that were acquired after re-probing of the same membrane. For fold change values of activated versus non-activated T cells, the ratio of intensities normalized to β-Actin was calculated. For statistical analysis, the fold change ratio was log2-transformed and subjected to one-way ANOVA (GraphPad Prism 8 software) analysis.

## IL-1β and IL-18 ELISA

Concentrations of cytokines in cell culture supernatants were determined by ELISA according to the manufacturer's instructions. The following kits were used: Human IL-1β ELISA Set II (BD Biosciences), Human Total IL-18 DuoSet ELISA (R&D Systems), Human Interferon-gamma DuoSet ELISA (R&D Systems), and Human IL-4 DuoSet ELISA (R&D Systems).

## Microscopy

FulTrac microinserts (Ibidi) were placed into 24-well µ-plates (Ibidi). The wells within the insert were prepared according to the manufacturer's instructions. Subsequently, 8 µl medium containing 2,000 cells and 1:50 propidium iodide (eBioscience) were put in each microinsert and allowed to settle down for 1 h. Treatment was started by the addition of the indicated stimulus in 2 µl volume. Images were acquired on a Leica DMi8-inverted microscope equipped with one of the following objectives: HC PL FLUOTAR L 25×/ 0.80 IMM and HC PL APO 63×/ 1.20 W CORR CS2.

## Quantification of PI-positive cells

Image segmentation of the T cells was performed using a custom-written *Definiens XD 2.0* script. Contours of individual cells were

determined using the transmitted light images (Channel1) that was acquired along with the fluorescence signal emitted by the PI-positive T cells (Channel2). As individual cells appear as dark, relatively round objects in Channel1, we applied the following strategy for their detection: After inversion of Channel1, we applied a 3D-Gaussian filter with a kernel size of $5 \times 5 \times 3$ pixels, then applied a second 3D-Gaussian filter, again with a kernel size of $5 \times 5 \times 3$ pixels. We finally subtracted the latter image from the previous one. This procedure resulted in a background-subtracted image. As a last step, we applied a global threshold and carried out segmentation using an algorithm implemented in the *Definiens XD 2.0* software platform. Briefly, the used so-called Multi-Threshold Segmentation algorithm splits the image domain and classifies the resulting image objects based on a defined pixel value threshold. The average fluorescence intensity corresponding to the PI level (Channel2) was then computed for each of the resulting segmented patterns. This procedure allowed extracting the total number as well as the number of PI-positive T cells in each frame. The fraction of PI-positive cells was then calculated for every image, and the average of PI-positive cells was calculated over a sliding window covering the following 30 min of each image. For quantification of MDMs in Fig EV2B, this approach was not applicable, as we could not quantify the total number of cells by identifying macrophages in an automated fashion due to their heterogenous morphology. Therefore, in this experiment, the PI signal of each image was normalized to the number of cells as visually determined at the start of the experiment.

### Knockout generation in primary human T cells

After isolation, CD4 T cells were incubated in medium containing IL-7 and IL-15 overnight. On the following day, T cells were washed once in PBS and $2 \times 10^6$ T cells were resuspended in 20 μl buffer P3 (Lonza). In parallel, CRISRP-Cas9-RNPs were assembled by annealing synthetic, chemically stabilized crRNA:tracrRNA pairs (synthego or IDT) at 95°C for 5 min and incubation at RT for 30 min. gRNAs were then mixed with recombinant NLS-Cas9 protein for 10 min at RT. For every gRNA, 100 pmol were combined with 40 pmol Cas9 protein. When targeting with several gRNAs, 40 pmol Cas9 was added for each 100 pmol of gRNA. In the case of CASP1-knockout clones (see below), a synthetic sgRNA (gRNA 1, Synthego) was used instead of crRNA:tracrRNA pairs. In this case, the annealing step was omitted.

Ribonucleoproteins were then mixed with the cell suspension and nucleofection was conducted (program EH100) on the X-unit of a 4D-nucleofector (Lonza; Seki & Rutz, 2018). After nucleofection, cells were collected from the nucleofection cuvettes with serum-free RPMI (w/o supplements) and transferred into 24-well plates. Cells were allowed to recover for 30–60 min in the incubator. Consecutively, T cells were activated with human T activator CD3/CD28 beads (bead:-cell ratio 1:5) in the presence of IL-7, IL-15, IL-2, and 2.5% human serum.

For the generation of single cell clones with all-allelic knockout for the gene of interest, T cells were subjected to minimal dilution cloning on day 2 of activation in round-bottom 96-well plates in medium containing IL-2 in the absence of feeder cells. In these experiments MACS-purified naïve CD4 T cells were used. Mono-clones were restimulated on day 8 by addition of 2000 CD3/CD28 beads/well. When colonies became visible after approximately 3 weeks, clones were collected and subjected to MiSeq analysis as described before (Schmid-Burgk *et al*, 2014). In order to obtain

sufficient cellular material, all clones generated that were all-allelic knockouts for a certain gene or wildtype were pooled. Wildtype clones from the same donor were pooled from different genome-targeting approaches. For the generation of knockout clones, only individual gRNAs were used for ASC (gRNA1), CASP1 (gRNA1), NLRP1 (gRNA1), and GSDMD (gRNA1), and a pair of gRNA was used for CARD8 (gRNA1 and gRNA2).

For polyclonally edited CD4 T-cell pools, pairs of gRNAs were used (CARD8 gRNA1 + gRNA2, ASC gRNA1 + gRNA2, CASP1 gRNA 2 + gRNA3, NLRP1 gRNA1 + gRNA2, GSDMD gRNA 1 + gRNA2). Subsequently, cells were expanded in IL-2 for 10–14 days, subsequently CD3/CD28 beads were removed and cells were switched to IL-7 and IL-15 containing medium for at least 4 days before experiments were conducted. No fresh cytokines were added on the day of the experiment. Polyclonal editing efficiency was confirmed by MiSeq, Sanger sequencing, or immunoblotting. Sequences of gRNAs are listed in the Table EV1.

### Knockout generation in THP-1 cells

THP-1 cells were gene targeted with RNPs. RNP assembly was performed as described above. For nucleofection (program FF100), 500,000 cells were resuspended in 20 μl nucleofection buffer P3. For ASC (gRNA1), CASP1 (gRNA1, sgRNA, annealing step omitted), and NLRP3 (gRNA 1), individual gRNAs were used, whereas for CARD8 (gRNA3 + gRNA4) two gRNAs were used. After nucleofection, cells were collected from the nucleofection cuvettes with serum-free RPMI (w/o supplements) and transferred into 24-well plates. Cells were allowed to recover for 30–60 min in the incubator. Subsequently, medium was added to 10% FCS (v/v) and THP-1 cells were rested for 48 h before cells were subjected to minimal dilution cloning. Knock-out clones were identified by MiSeq as described above. The parental cell line was used as wild-type control. Sequences of gRNAs are listed in the Table EV1.

### Analysis of a publicly available RNASeq dataset

The public RNASeq dataset GSE107011 containing 29 different immune cell types from 4 different health donors was used to quantify gene expression levels within T cell and monocyte subsets. The processed data containing TPM (transcripts per million) values were obtained from "ftp://ftp.ncbi.nlm.nih.gov/geo/series/GSE107nnn/ GSE107011/suppl/GSE107011_ProcProce_data_TPM.txt.gz". The Ensembl IDs were converted into gene symbols based on the gene annotation file from Ensembl release 98. The average values across different donors were used to represent the gene expression levels for each cell type. TPM values were trimmed to the range between $2^{-2}$ and $2^{12}$. The log2 transformation of the trimmed TPM values is shown in Fig 3A.

## Data availability

This study includes no data deposited in external repositories. The public RNASeq dataset re-analyzed in this study (Fig 3A) is available under GSE107011.

**Expanded View** for this article is available online.

## Acknowledgements

We kindly thank Claudia Ludwig and Andreas Wegerer (Gene Center, LMU) for great technical support, Joshua Kie from the BioSysM FACS Core Facility (Gene Center, LMU) for cell sorting, Russell Vance (UC Berkeley, USA) for providing us with the NeedleTox expression plasmid and Manuela Moldt and Karl-Peter Hopfner (Gene Center, LMU) for help producing the NeedleTox protein, Sabine Suppmann (MPI, Munich) for providing us with recombinant CSF-1 and NLS-Cas9, Raymund Buhman and Andreas Humpe for providing leukocyte reduction system chambers (Department of Transfusion Medicine, LMU). AL is supported by a fellowship of the Else Kröner-Forschungskolleg: Cancer Immunotherapy and the FöFoLe program of the LMU. This work was supported by grants from the ERC (ERC-2014-CoG − 647858 GENESIS) and funded by the Deutsche Forschungsgemeinschaft (DFG, German Research Foundation) TRR 237/A09 and SFB 1335/P015 to VH and SPP 1923 to OTK and VH. Open access funding enabled and organized by Projekt DEAL.

## Author contributions

Conceptualization: AL, SB, VH; Formal analysis: YC, CJ, MA; Investigation: AL, SB; Writing: VH with input from all authors; Resources: OTK, VH; Funding acquisition: OTK, VH; Supervision: VH.

## Conflict of interest

VH serves on the Scientific Advisory Board of Inflazome Ltd.

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
