## [Review Process File · The EMBO Journal]

CARD8 inflammasome signaling in human T cells

Andreas Linder, Stefan Bauernfried, Yiming Cheng, Manuel Albanese, Christophe Jung, Oliver Keppler, and Veit Hornung

DOI: 10.15252/emboj.2020105071

Corresponding author(s): Veit Hornung (hornung@genzentrum.lmu.de)

Review Timeline:

Submission Date:	24th Mar 20
Editorial Decision:	14th Apr 20
Revision Received:	20th Jun 20
Editorial Decision:	13th Jul 20
Revision Received:	17th Jul 20
Accepted:	22nd Jul 20

Editor: Karin Dumstrei

Transaction Report:

Dear Veit,

Thank you for submitting your manuscript to The EMBO Journal. Your study has now been seen by three referees.

As you can see from the comments below, the referees find the analysis interesting and suitable for consideration here. They raise some concerns regarding the in vivo significance of the reported findings. Some of the concerns can be addressed with text changes and expansion of the discussion, for some of the others it would be good to include additional data. I think it would be helpful to discuss the revisions further via phone or skype. Then we can also discuss timeline, what data you have on hand etc.

When preparing your letter of response to the referees' comments, please bear in mind that this will form part of the Review Process File, and will therefore be available online to the community. For more details on our Transparent Editorial Process, please visit our website:

<https://www.embopress.org/page/journal/14602075/authorguide#transparentprocess>

Thank you for the opportunity to consider your work for publication. I look forward to discussing your revisions further

with best wishes

Karin

Karin Dumstrei, PhD
Senior Editor
The EMBO Journal

- a point-by-point response to the referees' comments, with a detailed description of the changes made (as a word file).
- a word file of the manuscript text.
- individual production quality figure files (one file per figure)

- a complete author checklist, which you can download from our author guidelines (<https://www.embopress.org/page/journal/14602075/authorguide>).

- Expanded View files (replacing Supplementary Information)

Further information is available in our Guide For Authors:

The revision must be submitted online within 90 days; please click on the link below to submit the revision online before 13th Jul 2020.

Link Not Available

Referee #1:

Comments on the study from Linder et al.,

In their study Linder et al, elegantly show that naïve human T cells express a functional CARD8 inflammasome that is activated by the valbopro compound. Importantly, T cells seem to be insensitive to other inflammasome triggers (e.g. flatox for NLRC4, Nlgericin for NLRP3). In addition, they describe that activated T cells become unresponsive to val boro pro -activated CARD8, suggesting that an additional layer of the CARD8 inflammasome regulation is acquired upon T cell activation/maturation.

Overall, the study is very well designed and executed, and it opens a lot of new questions regarding the function of CARD8-induced pyroptosis in naïve T cells. Does it control maturation of T cells, positive or negative selection (as TCR engagement leads to CARD8 inhibition), an easy/naïve bet would be an involvement in negative selection.

Another standing question, is also whether CARD8 evolved to be a microbial sensor in T cells. If so, evolutionary T-cell adapted virus might help answering such response. The authors suggest that CARD8 might be a putative HIV sensor. Yet, HIV-human interactions are only recent in regard of CARD8 expression in T cells, which does not imply that CARD8 might be a HIV direct or indirect sensor. Could the authors speculate regarding CARD8 and NLRP1 activation mechanisms about the type (not the identity) of HIV-derived effector that could promote theoretically CARD8 response in T cells?

The authors discuss the physiological relevance of CARD8 expression, but two studies reported that CARD8 loss of function leads to increased NLRP3 inflammasome response in myeloid cells:

- Did the authors find such results in Nigericin +/- MCC950-treated CARD8-deficient T cells ?

doi: 10.1172/JCI98642

doi: 10.1186/ar4483.

- The authors show that T cells undergo pyroptosis upon valproic acid exposure but due to the lack of IL1b/IL18 and IL1alpha expression, there is no IL-1-derived cytokine release. What about other alarmins such as HMGB1 ? And other IL-1-derived cytokine/alarmins (e.g. IL-36 alpha beta and gamma, IL38...)

- Caspase-4 seems to be strongly expressed in T cells. What about the non-canonical inflammasome ?

- The authors refer to AIM2- or Irf1-induced T cell pyroptosis upon HIV infection. Did they test this ?

- Nlrp1 is expressed in T cells. Is anthrax factor triggering Nlrp1 and/or CARD8-dependent pyroptosis as anthrax toxins were found to alter directly T cell-based response.

Overall this is a beautiful study which opens a full land of novel questions in the field of understanding inflammasome biology in T cells.

Referee #2:

In this manuscript, the authors identify an important function of the newly described CARD8 inflammasome in mediating pyroptosis of T cells. The work is interesting and carefully performed. I am strongly in favor of publication.

Minor comments

line 49 - the number of human NAIP paralogs is not well established. To be safe, perhaps just say "...humans appear to encode only one NAIP..." or "...humans encode only one well-characterized NAIP..."

line 129 - it is not entirely clear to me whether it has been shown that increased osmotic pressure is the cause of pyroptosis. Perhaps a reference for this statement would be appropriate? The authors are surely aware there are dissenting views on this (e.g., PMID 30796192), so some caution may be warranted. Indeed, while I am certainly not a biophysicist, it seems to me that large membrane pores would dissipate osmotic pressure, not cause it.

line 210 - Although the T cells don't appear to respond to NeedleTox (as expected given lack of NAIP/NLRC4), is it clear that T cells express the ANTXR necessary for the uptake of NeedleTox? Expression data, or a demonstration that the cells are susceptible to Edema Factor/PA would address this.

Referee #3:

Review of the manuscript 'CARD8 inflammasome signaling in human T cells' by Dr. Veit Hornung and colleagues.

In this manuscript, the authors demonstrate that the human-specific sensor CARD8 triggers gasdermin D (GSDMD)-dependent pyroptosis in resting/non-activated CD4⁺ T cells. Using a combinational approach employing specific apoptosis (ABT737) vs. pyroptosis/DPP (ValboroPro) inhibitors and CRISPR-Cas9 technology to genetically ablate CARD8 and key connected pathways components they aim to demonstrate that the inhibition of the CARD8 'restrainer' DPP9 causes lytic cell death without the involvement of other inflammasomes. Thus, they identify a DPP9-CARD8-caspase-1-GSDMD-axis that was thought to be only operative in myeloid cells as a key pyroptosis controller in T lymphocytes.

The major strength of this work lays in the authors technically careful and convincing approach to demonstrate the existence of this novel DPP9-CARD8-caspase-1-GSDMD-axis in human CD4⁺ T cells. Thus, I fully support their interpretation that under their ex vivo conditions tested, human CD4⁺ T cells are capable of undergoing pyroptosis in a CARD8 pathway-dependent manner. This would also be commensurate with recent findings that T cells engage innate immune sensors previously thought only present in immune cells of the myeloid lineage.

A major weakness of the work, however, lays in the fact that the authors leave us uninformed about the in vivo significance of their observation. The fact that CARD8 is not expressed in mice precludes direct assessment in a small animal model, but there may be other ways to begin probing this pathway in vivo (see below).

Further, that large majority of functional data focus on CD4⁺ T cells (except for data in Figure 1), thus, the authors should reflect this in the title and throughout the text and state 'in CD4⁺ T cells' throughout. Finally, the activation conditions of the T cells seem in part inconsistent and I would suggest considering 1-2 control conditions during key experiments (Figure 1).

Suggestions/queries:

1. It is very difficult to follow the exact T cell activation protocols for the different experiments. Are all cells isolated rested for 3 days before usage (why)? Is there IL-2 during stimulation added or not? T cell activation in Figure 1 seems to be performed without cytokines (IL-2) and in 2.5% human serum (this is very low, usually 5% and above are added). Also, the number of cells cultured in 96-well plates is unusually high (in our hands, under the conditions listed, the cells 'crash' overnight). CRISPR-Cas9-derived T cells have been 'generated' in the presence of cytokines and thus experienced signals derived from these important growth/survival factors. How do the data compare when culture media with higher HS and IL-2 addition are utilized? I noticed that non-treated T cells in Figure 2A look already quite 'unhealthy' after 16 hrs in culture - this is not normal and could indeed be due to unfavorable conditions.
2. How do the expression levels of proteins in resting and activated T cells when normalized against beta-actin levels (Figure 5B).
3. Since DPP9 is also impacting on NLRP1 as well as NF- κ B activity (which controls CD4⁺ T cell death) can the authors include monitor activity of these two components in the DPP9 KO cells (there seems to be a trend in reduction of pyroptosis in the NLRP1 KO cells - Figure 3D).
4. An experiment to assess for the in vivo impact of CARD8-loss may be to inject CARD8 CRISPR-ed human T cells into an NSG mouse strain. This would also have the advantage that the initial expansion phase of injected T cells is based on lymphopenic expansion and should be stimulation independent, whilst there is then a second phase of T cell stimulation (mostly in the intestine).
5. Most work describing the presence of a function NLRP3 inflammasome in human CD4⁺ T cells

indicate that NLRP3-driven IL-1beta production becomes only solidly measurable when assessing cells from patients with CAPS (Arbore et al, Science 2016, cited) or arthritis (et al, - Li et al. Cell Metabolism 2019, not cited). Could the authors source T cells from patients with arthritis and assess the DPP9-CARD8-caspase-1-GSDMD-axis/deviations in their pathway in those cells?

6. Although the authors interrogated T cells with CARD8 ablation, it may be interesting to assess T cells from patients with CARD8 deficiency (Mao et al., JCI 2018, not cited), if they can be sourced, that it. As such cells have been 'born' without CARD8 they could be a great source to evaluate for changes in pyroptosis (that me become better apparent over the life time of a T cell) and possibly for their TCR repertoire (as the authors suggested that this may be altered).

Point-by-point response Linder et al.

Foremost, we would like to thank the reviewers for their valuable time assessing our manuscript in great detail. We now present a thoroughly revised manuscript addressing most of the critical points raised by the reviewers. We feel that by addressing these comments and suggestions, the manuscript has much improved, and we kindly want to thank the reviewers for taking their time to review our manuscript and for providing valuable comments and suggestions.

Referee #1:

Comments on the study from Linder et al.,

In their study Linder et al, elegantly show that naïve human T cells express a functional CARD8 inflammasome that is activated by the valboropro compound. Importantly, T cells seem to be insensitive to other inflammasome triggers (e.g. flatox for NLRC4, Nigericin for NLRP3). In addition, they describe that activated T cells become unresponsive to val boro pro -activated CARD8, suggesting that an additional layer of the CARD8 inflammasome regulation is acquired upon T cell activation/maturation.

Overall, the study is very well designed and executed, and it opens a lot of new questions regarding the function of CARD8-induced pyroptosis in naïve T cells. Does it control maturation of T cells, positive or negative selection (as TCR engagement leads to CARD8 inhibition), an easy/naïve but would be an involvement in negative selection.

Another standing question, is also whether CARD8 evolved to be a microbial sensor in T cells. If so, evolutionary T-cell adapted virus might help answering such response. The authors suggest that CARD8 might be a putative HIV sensor. Yet, HIV-human interactions are only recent in regard of CARD8 expression in T cells, which does not imply that CARD8 might be a HIV direct or indirect sensor. Could the authors speculate regarding CARD8 and NLRP1 activation mechanisms about the type (not the identity) of HIV-derived effector that could promote theoretically CARD8 response in T cells?

The nature of the naturally occurring signal that activates CARD8 is not known. Inferring from NLRP1 which seems to have evolved to act as a decoy and to detect enzymatic activity of pathogens one could speculate on an analogous mechanism for CARD8. Yet, as we have not investigated whether HIV can trigger CARD8-dependent pyroptosis we are hesitant to speculate on the nature of HIV-derived signals that theoretically could act on CARD8.

The authors discuss the physiological relevance of CARD8 expression, but two studies reported that CARD8 loss of function leads to increased NLRP3 inflammasome response in myeloid cells:

- Did the authors find such results in Nigericin +/- MCC950-treated CARD8-deficient T cells ?

doi: 10.1172/JCI98642

doi: 10.1186/ar4483.

We thank the reviewer for pointing out these studies. To address the role of CARD8 in myeloid cells, we generated CARD8- and NLRP3-deficient THP1 cells and subjected these cells to either VbP or Nigericin stimulation. Doing so, we could not observe a negative regulatory effect of CARD8 on NLRP3 activation, while CARD8 was indeed functional in THP1 cells as previously reported by the lab of Dan Bachovchin (PMID 29967349). We now included these data in our manuscript (Fig. EV3). Additionally, when gene-targeting CARD8 in primary monocyte-derived macrophages (and ASC as a control) in a preliminary experiment with two donors, we again could not observe a phenotype that would suggest a negative regulatory role for CARD8 on NLRP3 activation (Additional Fig. 1A - C attached to this point-to-point reply)

- The authors show that T cells undergo pyroptosis upon val boron exposure but due to the lack of IL1 β /IL18 and IL1 α expression, there is no IL-1-derived cytokine release. What about other alarmins such as HMGB1? And other IL-1-derived cytokine/alarmins (e.g. IL-36 α , IL-36 β and IL-38...)

The reviewer raises an important point. While CARD8-stimulated T cells clearly undergo a lytic cell death, we were not able to detect any IL-1-related cytokines in their supernatant. (In addition to IL-1 β , we now also included the measurement of IL-18 in our manuscript, Fig. 1C). This is well in line with the fact that these cytokines are not expressed in T cells (see also additional Fig. 1D). In addition, we also failed to detect HMGB1 being released from pyroptotic T cells via western blot or mass spectrometry (data not shown) although it is abundantly expressed in T cells. As such, we currently cannot provide any information on the pro-inflammatory nature of this type of cell death. In future studies, we plan to conduct a series of functional assays with the supernatant of CARD8-stimulated T cells to identify potential DAMPs or alarmins. Yet, we hope that the reviewer agrees that it would be beyond the scope of this study to include such a set of experiments at this point.

- Caspase-4 seems to be strongly expressed in T cells. What about the non-canonical inflammasome?
- The authors refer to AIM2- or Ifi16-induced T cell pyroptosis upon HIV infection. Did they test this?

Up to now, we have not extensively investigated the role of the non-canonical inflammasome in human T cells. In our understanding, the non-canonical inflammasome serves to detect cell wall components of gram-negative bacteria that escape from endosomes in epithelial or phagocytic cells. To our knowledge there is no gram-negative bacterium that infects T cells. The non-canonical inflammasome is additionally regulated on the level of guanylate binding proteins (GBPs), which require interferon-induced genes expression to become functional. Our preliminary data suggests, that transfecting LPS readily induces a lytic form of cell death in monocyte-derived macrophages but not in T cells, irrespective of type I interferon priming (Additional Fig. 2A). So far, we have not had the opportunity to assess pyroptosis upon HIV infection experimentally. With the availability of human knock-out T cells, these findings should be reevaluated in the future.

- Nlrp1 is expressed in T cells. Is anthrax factor triggering Nlrp1 and/or CARD8-dependent pyroptosis as anthrax toxins were found to alter directly T cell-based response.

While Anthrax Lethal factor is a known activator of certain murine Nlrp1b alleles, it fails to activate human NLRP1 or CARD8 due to the lack of the cleavage site in the human orthologue. This is well documented in the literature. Moreover, an earlier report did not observe an impact on T cell viability upon treatment with either of the anthrax toxins (PMID 15699068).

Overall this is a beautiful study which opens a full land of novel questions in the field of understanding inflammasome biology in T cells.

We thank the reviewer for her/his appreciation of our work.

Referee #2:

In this manuscript, the authors identify an important function of the newly described CARD8 inflammasome in mediating pyroptosis of T cells. The work is interesting and carefully performed. I am strongly in favor of publication.

Minor comments

line 49 - the number of human NAIP paralogs is not well established. To be safe, perhaps just say "...humans appear to encode only one NAIP..." or "...humans encode only one well-characterized NAIP..."

We thank the reviewer for her/his insight and edited the text as suggested (line 49 and line 129).

line 129 - it is not entirely clear to me whether it has been shown that increased osmotic pressure is the cause of pyroptosis. Perhaps a reference for this statement would be appropriate? The authors are surely aware there are dissenting views on this (e.g., PMID 30796192), so some caution may be warranted. Indeed, while I am certainly not a biophysicist, it seems to me that large membrane pores would dissipate osmotic pressure, not cause it.

We agree with the reviewer that our phrasing was too vague and imprecise. We therefore edited the text accordingly and now also refer to Davis et al. PNAS 2019.

line 210 - Although the T cells don't appear to respond to NeedleTox (as expected given lack of NAIP/NLRC4), is it clear that T cells express the ANTXR necessary for the uptake of NeedleTox? Expression data, or a demonstration that the cells are susceptible to Edema Factor/PA would address this.

While our experimental data formally cannot rule out that the unresponsiveness of T cells to NeedleTox is due to inefficient protective antigen mediated uptake of the toxin, the ANTXR2 gene is expressed in T cells to a similar extent as in monocytes, while the ANTXR1 seems not to be expressed in either of the two classes of cells. A respective expression profile is provided in the attached figure (Additional Fig. 1D). That T cells actually are responsive to the anthrax toxins that rely on PA-mediated uptake has been shown earlier (PMID 15699068). Nevertheless, we now additionally point out that we cannot rule out that T cells do not take up NeedleTox and thus appear to be non-functional for the NAIP/NLRC4 inflammasome (line 249 -252)

Referee #3:

Review of the manuscript 'CARD8 inflammasome signaling in human T cells' by Dr. Veit Hornung and colleagues.

In this manuscript, the authors demonstrate that the human-specific sensor CARD8 triggers gasdermin D (GSDMD)-dependent pyroptosis in resting/non-activated CD4⁺ T cells. Using a combinational approach employing specific apoptosis (ABT737) vs. pyroptosis/DPP (ValboroPro) inhibitors and CRISPR-Cas9 technology to genetically ablate CARD8 and key connected pathways components they aim to demonstrate that the inhibition of the CARD8 'restrainer' DPP9 causes lytic cell death without the involvement of other inflammasomes. Thus, they identify a DPP9-CARD8-caspase-1-GSDMD-axis that was thought to be only operative in myeloid cells as a key pyroptosis controller in T lymphocytes.

The major strength of this work lays in the authors technically careful and convincing approach to demonstrate the existence of this novel DPP9-CARD8-caspase-1-GSDMD-axis in human CD4⁺ T cells. Thus, I fully support their interpretation that under their ex vivo conditions tested, human CD4⁺ T cells are capable of undergoing pyroptosis in a CARD8 pathway-dependent manner. This would also be commensurate with recent findings that T cells engage innate immune sensors previously thought only present in immune cells of the myeloid lineage.

A major weakness of the work, however, lays in the fact that the authors leave us uninformed about the in vivo significance of their observation. The fact that CARD8 is not expressed in mice precludes direct assessment in a small animal model, but there may be other ways to begin probing this pathway

in vivo (see below).

Further, that large majority of functional data focus on CD4⁺ T cells (except for data in Figure 1), thus, the authors should reflect this in the title and throughout the text and state 'in CD4⁺ T cells' throughout. Finally, the activation conditions of the T cells seem in part inconsistent and I would suggest considering 1-2 control conditions during key experiments (Figure 1).

Suggestions/queries:

1. It is very difficult to follow the exact T cell activation protocols for the different experiments. Are all cells isolated rested for 3 days before usage (why)? Is there IL-2 during stimulation added or not? T cell activation in Figure 1 seems to be performed without cytokines (IL-2) and in 2.5% human serum (this is very low, usually 5% and above are added). Also, the number of cells cultured in 96-well plates is unusually high (in our hands, under the conditions listed, the cells 'crash' overnight). CRISPR-Cas9-derived T cells have been 'generated' in the presence of cytokines and thus experienced signals derived from these important growth/survival factors. How do the data compare when culture media with higher HS and IL-2 addition are utilized? I noticed that non-treated T cell in Figure 2A looks already quite 'unhealthy' after 16 hrs in culture - this is not normal and could indeed be due to unfavorable conditions.

Whenever monocytes were differentiated into macrophages as control conditions, T cells were kept resting in the presence of IL-7/IL-15 until the monocyte differentiation was completed (Fig. 1, Fig. 2 A and B). Viability of T cells was monitored by trypan-blue exclusion. In some other experiments T cells were directly stimulated on the day of extraction or rested overnight in the absence of cytokines (Fig. 2C, Fig. EV2C).

We could not observe a decrease in susceptibility towards VbP-induced cell death, no matter whether the T cells were stimulated on the day of extraction or kept for several days in the presence of IL-7/IL-15. Moreover, the serum-content (2,5 % human serum vs 5% vs 10% FCS) did not impact on susceptibility. Addition of IL-2 to resting or activated T cells during VbP-treatment nor preculturing the cells in the presence of IL-2 did not impact on susceptibility to VbP-induced cell death. A respective experiment illustrating this is included in the attached figure (Additional Fig. 2B and C). Moreover, we also added a note in the methods section.

In Fig. 2A, the T cells were kept in micro inserts in a total volume of 10 μ l only, which could potentially impact on cell viability. However, based on PI uptake, the T cells did not die significantly in the untreated conditions. A quantification of the PI uptake is now included in Fig. EV2B.

2. How do the expression levels of proteins in resting and activated T cells when normalized against beta-actin levels (Figure 5B).

The reviewer is right that the previous immunoblots displayed higher b-actin signals, which should be taken into account when making a statement towards the expression levels of individual proteins (even though these blots were normalized for protein amounts). To provide more quantifiable results, we now included the quantification of immunoblots from several additional experiments (9 donors in total, new Fig. 5C)). In addition, we replaced Fig. 5B with a blot that better represents the overall changes in expression levels when taking into account all donors analyzed. While these experiments show a great donor variability, we found that while the full-length variants of CARD8 tends to be upregulated (and caspase-1 and GSDMD remain the same), the portion of auto-cleaved CARD8 slightly decreases. This could point towards a mechanistic explanation as to why T cells become so resistant to VbP after activation, which we now also discuss in our manuscript.

3. Since DPP9 is also impacting on NLRP1 as well as NF- κ B activity (which controls CD4⁺ T cell death) can the authors include monitor activity of these two components in the DPP9 KO cells (there seems to be a trend in reduction of pyroptosis in the NLRP1 KO cells - Figure 3D).

We are not sure what the reviewer means by impacting on NF- κ B activity, since we neither directly nor indirectly addressed NF- κ B activation in our study. Moreover, we are not aware of any connection of DPP9 and NF- κ B activation. Of note, we could not find any evidence that NLRP1 deficiency (or ASC-deficiency, which acts downstream of NLRP1) protects T cells from VbP-induced cell death, based on both LDH-release (Fig. 3D) and microscopy of KO-clones (Fig. EV4C).

4. An experiment to assess for the *in vivo* impact of CARD8-loss may be to inject CARD8 CRISPR-ed human T cells into an NSG mouse strain. This would also have the advantage that the initial expansion phase of injected T cells is based on lymphopenic expansion and should be stimulation independent, whilst there is then a second phase of T cell stimulation (mostly in the intestine).

We would be delighted to investigate the T cell intrinsic role of CARD8 *in vivo* and we indeed submitted a grant application in this direction. As pointed out by the referee, this requires the establishment of complex humanized mouse models, which are currently not established in our laboratory. We hope that the reviewer agrees that we consider it to be beyond the scope of this study to address this in this current manuscript.

5. Most work describing the presence of a function NLRP3 inflammasome in human CD4⁺ T cells indicate that NLRP3-driven IL-1 β production becomes only solidly measurable when assessing cells from patients with CAPS (Arbore et al, Science 2016, cited) or arthritis (et al, - Li et al. Cell Metabolism 2019, not cited). Could the authors source T cells from patients with arthritis and assess the DPP9-CARD8-caspase-1-GSDMD-axis/deviations in their pathway in those cells?

Unfortunately, cells from such patients are currently not available to us, apart from the fact that investigating such cells would require an additional ethics vote. We agree with the referee, that our data does not rule out that T cells can acquire expression of IL-1-related cytokines under certain circumstances and we additionally point this out in the discussion section (line 481). As mentioned above, further efforts, including the analysis pathological T cells *ex vivo*, could be instructive in future studies.

6. Although the authors interrogated T cells with CARD8 ablation, it may be interesting to assess T cells from patients with CARD8 deficiency (Mao et al., JCI 2018, not cited), if they can be sourced, that it. As such cells have been 'born' without CARD8 they could be a great source to evaluate for changes in pyroptosis (that may become better apparent over the life time of a T cell) and possibly for their TCR repertoire (as the authors suggested that this may be altered).

Unfortunately, cells from such patients are not available to us (please see also comment above). We inquired with our colleagues from the Department of Pediatrics (LMU Munich), who established one of the largest cohorts of patients with early onset IBD (example study with this cohort: PMID 19890111). However, they did not have any patients with such a mutation.

Instead, we investigated the role of CARD8 in negatively regulating the NLRP3 inflammasome, a concept that is mainly based on co-overexpression studies in the HEK293 system. In THP1 cells and primary macrophages we could not find evidence for CARD8 impacting on NLRP3 activation. The data is now included in Fig. EV2 and additional Fig. 2 A-C.

In light of the only recently established concept, that CARD8 can act as an inflammasome-sensor itself, the phenotype reported by Mao et al. could also be explained by a gain-of-function rather than a loss-of-function mutation of CARD8. Reinvestigating this and other variants of CARD8 that have been proposed to lead to a loss-of-function warrant reinvestigation in future studies in light of the novel function of CARD8.

Additional changes:

In addition to the changes in response to the referees' comments, we included a quantification of the PI positive cells of the movies EV1A-H. There is now a new Fig. EV2B, the former Fig. EV2B is now Fig. EV2C and the former Fig. EV2C is now Fig. EV2D. This analysis had not been included in the original manuscript due to technical difficulties in automating the detection of macrophages. Now this problem could be overcome and we consider this figure valuable in terms of visualizing the kinetics of the experiment. A description of gene-targeting and cell culture of THP-1 cells was added to the methods section.

Of note, as opposed to the original figures, we now do not assign specific isoforms to the bands that are detected by the CARD8 antibodies (e.g. in Fig. 3C) in the revised manuscript. We know that these bands are CARD8-specific, given the fact that their expression is completely abolished in both T cells and THP-1 cells when gene targeting CARD8 with CRISPR-Cas9. Yet, since these bands do not precisely match the predicted sizes of annotated CARD8 isoforms that we had initially assumed, we now opted to not assign specific isoforms to them.

Overview

Figure	Change in the revised manuscript
Fig. 1A	unchanged
Fig. 1B	unchanged
Fig. 1C	New data depicting IL-18 production
Fig. 2A	unchanged
Fig. 2B	unchanged
Fig. 2C	unchanged
Fig. 3A	Unit "log ₂ of TPM" added to the legend
Fig. 3B	Annotation of specific CARD8 isoforms removed
Fig. 3C	Annotation of specific CARD8 isoforms removed
Fig. 3D	Layout change: width of the bars decreased, legend adjusted
Fig. 4A	unchanged
Fig. 4B	unchanged
Fig. 4C	unchanged
Fig. 5A	Layout change: width of the bars decreased
Fig. 5B	Figure exchanged for a more representative western blot, NLRP1 expression removed as not relevant and not available for all immunoblots quantified in figure 5C
Fig. 5C	New figure showing quantification of immunoblots in figure 5B and additional immunoblots
Fig. EV1A	Now individual data points included to meet EMBO journal guidelines, layout change: new coloring, width of bars adjusted
Fig. EV2A	unchanged
Fig. EV2B	New figure showing quantification of data shown in figure 2A and Movie EV1
Fig. EV2C	Formerly Figure EV2B, otherwise unchanged
Fig. EV2D	Formerly Figure EV2C, layout out change: width of bars and legend adjusted
Fig. EV3A	New figure showing immunoblotting of THP-1 KO cell lines
Fig. EV3B	New figure showing LDH release of THP-1 KO cell lines
Fig. EV3C	New figure showing IL-1 β production of THP-1 KO cell lines
Fig. EV4A	Formerly Figure EV3A, otherwise unchanged
Fig. EV4B	Formerly Figure EV3B, otherwise unchanged
Fig. EV4C	Formerly Figure EV3C, layout change: legend adjusted, size of fonts adjusted
Additional Fig. 1A	New figure showing LDH release of CRISPR-edited primary monocyte-

	derived macrophages, not part of the manuscript
Additional Fig. 1B	New figure showing IL-1 β production of CRISPR-edited primary monocyte-derived macrophages, not part of the manuscript
Additional Fig. 1C	New figure showing IL-6 production of CRISPR-edited primary monocyte-derived macrophages, not part of the manuscript
Additional Fig. 1D	New figure showing transcription of additional genes based on the data set shown in Figure 3A, not part of the manuscript
Additional Fig. 2A	New figure showing LDH release upon LPS-transfection of MDMs and T cells, not part of the manuscript
Additional Fig. 2B	New figure showing LDH release of T cells cultured in different media, not part of the manuscript
Additional Fig. 2C	New figure showing LDH release of T cells cultured, activated and stimulated in presence or absence of IL-2 , not part of the manuscript

[Figures for referees not shown.]

Dear Veit,

Thank you for submitting your revised manuscript to The EMBO Journal. Your study has now been re-reviewed by referee #1 and as you can see below, the referee appreciates the introduced changes and supports publication here.

I am very happy to let you know that we will accept your manuscript for publication here. Before sending you the formal acceptance letter there are just a few editorial things to sort out.

- we are missing 3-5 keywords
- we don't allow data not shown (page 17). You can either re-phrase or show the data
- Synopsis image looks good can you please make sure the size is OK. Should be 550 wide by [200-400] high (pixels)
- We require a Data availability section. As far as I can see no data is generated that needs to be deposited in a database and if correct please state: This study includes no data deposited in external repositories
- There are two tables please make sure there is a call out to them in the main MS file.
- The legend for each movie should be zipped with the movie. I think it is also easiest to re-label them with an individual figure number like movie EV4, movie EV5 etc. Please also correct call out in text.
- I have looked at the source data and it looks good - please upload the files as one file per figure.
- For figure 2A will you please double check that in the Needle Tox panels that timepoints 4 and 8 hr as well as the timepoints 12 and 16 hrs are not duplicates.
- Scale bars are missing in EV2A and very difficult to see in EV4C
- I have asked our publisher to do their pre-publication checks on the paper. They will send me the file within the next few days. Please wait to upload the revised version until you have received their comments.

That should be all - let me know if we need to discuss anything further. You can use the link below to upload the revised version.

Best Karin

Karin Dumstrei, PhD
Senior Editor
The EMBO Journal

- a point-by-point response to the referees' comments, with a detailed description of the changes made (as a word file).

- a word file of the manuscript text.

- individual production quality figure files (one file per figure)

- a complete author checklist, which you can download from our author guidelines (<https://www.embopress.org/page/journal/14602075/authorguide>).

- Expanded View files (replacing Supplementary Information)

Further information is available in our Guide For Authors:

The revision must be submitted online within 90 days; please click on the link below to submit the revision online before 11th Oct 2020.

Referee #1:

The authors have clarified all my questions.

I am happy to support their study for publication in EMBO Journal

Point-by-point response to the editor

Dear Dr. Dumstrei, we thank you for *in principle* accepting our manuscript for publication in EMBO Journal. We also thank the reviewers for the favorable evaluation of our revised manuscript.

We have addressed the suggested changes as follows:

- we are missing 3-5 keywords

We would like to include the following key words:

CARD8, pyroptosis, T cell, inflammasome, Val-boroPro,

The key words were also included in the manuscript.

- we don't allow data not shown (page 17). You can either re-phrase or show the data

The text was rephrased accordingly.

- Synopsis image looks good can you please make sure the size is OK. Should be 550 wide by [200-400] high (pixels)

The synopsis image was adjusted accordingly.

- We require a Data availability section. As far as I can see no data is generated that needs to be deposited in a database and if correct please state: This study includes no data deposited in external repositories

A data availability section was included in the manuscript at the end of the methods section.

- There are two tables please make sure there is a call out to them in the main MS file.

The tools and reagents table is now being called out in the manuscript at the beginning of the methods section. The EV table 1 (containing gRNA sequences) is called out in the methods section, where the genome editing is described.

- The legend for each movie should be zipped with the movie. I think it is also easiest to re-label them with an individual figure number like movie EV4, movie EV5 etc. Please also correct call out in text.

The names of the movies and call out in the text were changed in the manuscript as suggested. A separate word file "movie legends" was generated. This file was zipped with each movie.

- I have looked at the source data and it looks good - please upload the files as one file per figure.

Per figure, we uploaded one zipped source data PDF-file containing uncropped western blot images and one excel files containing raw data used to generate the figure.

- For figure 2A will you please double check that in the Needle Tox panels that timepoints 4 and 8 hr as well as the timepoints 12 and 16 hrs are not duplicates.

We took a thorough look at these images again, also at the versions before the vector graphic was transformed to a pixel graphic (to reduce image size) and we are sure that the images are not duplicates, based on both timestamp and movement of individual cells (one can see individual cells crawling around in the background).

- Scale bars are missing in EV2A and very difficult to see in EV4C

Scale bars were changed to 100 μm (Fig. EV2A) and 50 μm (Fig EV4C) respectively to increase visibility. The figure legends were adjusted accordingly.

- I have asked our publisher to do their pre-publication checks on the paper. They will send me the file within the next few days. Please wait to upload the revised version until you have received their comments.

The changes proposed by the publisher have all been incorporated in the revised version of the manuscript.

We hope that we have now addressed all the suggested editorial changes and that the manuscript is now ready for publication.

Dear Veit,

Thanks for sending me the revised manuscript. All looks good!

I am therefore very happy to accept the manuscript for publication here.

Congratulations on a nice paper!

with best wishes

Karin

Karin Dumstrei, PhD
Senior Editor
The EMBO Journal

Please note that it is EMBO Journal policy for the transcript of the editorial process (containing referee reports and your response letter) to be published as an online supplement to each paper. If you do NOT want this, you will need to inform the Editorial Office via email immediately. More information is available here: http://emboj.embopress.org/about#Transparent_Process

Your manuscript will be processed for publication in the journal by EMBO Press. Manuscripts in the PDF and electronic editions of The EMBO Journal will be copy edited, and you will be provided with page proofs prior to publication. Please note that supplementary information is not included in the proofs.

Should you be planning a Press Release on your article, please get in contact with embojournal@wiley.com as early as possible, in order to coordinate publication and release dates.

If you have any questions, please do not hesitate to call or email the Editorial Office. Thank you for your contribution to The EMBO Journal.

** Click here to be directed to your login page: <http://emboj.msubmit.net>

Corresponding Author Name: Veit Hornung

Journal Submitted to: The EMBO Journal

Manuscript Number: EMBOJ-2020-105071